# FlowSeg: Dynamic Semantic Guidance for LLM-Conditioned Segmentation

Zekang Zhang [1]  Guangyu Gao [1]  Youyun Tang [2]  ChengJing Wu [2]  Xiaochao Qu [2]  Chi Harold Liu [1]
Jianbo Jiao [3]  Yunchao Wei [4 5]  Luoqi Liu [2]  Ting Liu [2]

## Abstract

LLM-conditioned segmentation has recently advanced rapidly by coupling large language models with iterative mask generation frameworks. However, we identify a persistent failure mode in current query-based *propose-then-select* pipelines. Although high-quality mask candidates are often generated, the final prediction may fail to match the given linguistic condition. This failure arises because language semantics are typically used as static prompts or post-hoc matching signals, rather than participating in the iterative mask generation process. Through systematic analysis, we show that many errors stem from *semantic misalignment* rather than poor mask quality. To address this issue, we propose **FlowSeg**, which introduces *dynamic semantic guidance* via a *bidirectional semantic flow* between intermediate decoding states and LLM-derived condition embeddings throughout the generation process. Language conditions actively guide mask refinement at each stage, while condition embeddings are progressively updated by emerging visual evidence. This design yields semantically grounded mask representations and visually aligned language conditions, enabling more reliable matching. We further incorporate a lightweight boundary-aware refinement to selectively enhance uncertain regions without perturbing confident interiors. Extensive experiments on referring expression segmentation and reasoning segmentation tasks demonstrate that FlowSeg consistently improves language–mask alignment and achieves state-of-the-art performance. Project page: https://zkzhang98.github.io/FlowSeg_page.

[1]School of Computer Science, Beijing Institute of Technology [2]Meitu.inc [3]School of Computer Science, University of Birmingham [4]WEI Lab, Institute of Information Science, Beijing Jiaotong University [5]Beijing Key Laboratory of Advanced Information Science and Network. Correspondence to: Luoqi Liu <llq5@meitu.com>, Ting Liu <lt@meitu.com>.

*Proceedings of the 43rd International Conference on Machine Learning*, Seoul, South Korea. PMLR 306, 2026. Copyright 2026 by the author(s).

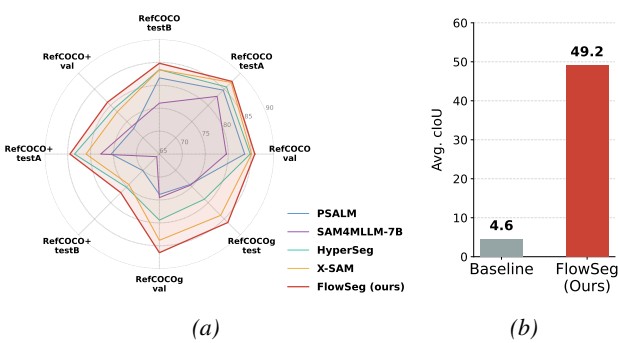

*Figure 1.* **(a)** Comparison with state-of-the-art methods on standard benchmarks. **(b)** Our method significantly improves performance on hard cases where the prior work fails ($cIoU < 0.5$).

## 1. Introduction

**LLM-conditioned segmentation** refers to segmenting image regions specified by natural-language conditions (Kazemzadeh et al., 2014; Mao et al., 2016), where large language models (LLMs) provide rich semantic representations to support open-ended instruction following and compositional understanding (Lai et al., 2024). Driven by recent progress in LLM-centric vision–language modeling, a series of methods have achieved strong performance by coupling LLMs with pixel-level segmentation modules (e.g., SAM-style predictors or query-based mask decoders) to output accurate masks under diverse prompts (Lai et al., 2024; Chen et al., 2024; Zhang et al., 2024b; Wei et al., 2024; Yuan et al., 2025; Wang et al., 2025).

Despite these advances, a family of high-performing LLM-conditioned segmentation systems has converged on a *propose-then-select* paradigm within **query-based** frameworks (Zhang et al., 2024b; Wei et al., 2024; Yuan et al., 2025; Wang et al., 2025). In this design, a set of learnable queries iteratively decode candidate masks from visual features, and the final prediction is obtained by matching these candidates with language condition embeddings via similarity scoring. By decoupling high-resolution mask decoding into specialized visual queries, this architecture enables fine-grained multi-scale interactions and has emerged as the leading design in LLM-conditioned segmentation. Yet we argue it introduces a subtle but systematic limitation that remains underexplored.

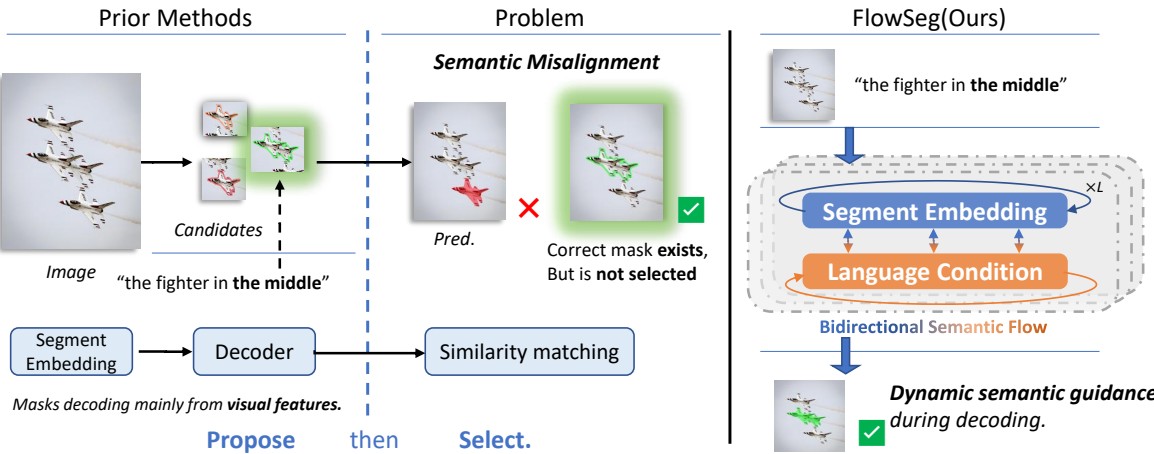

*Figure 2.* Motivation of FlowSeg. Existing query-based *propose-then-select* pipelines often generate accurate mask candidates but fail to select the one that matches the linguistic condition, due to limited semantic participation during iterative mask generation. Our proposed FlowSeg enables language conditions to integrate with and guide mask generation at each decoding stage, while allowing the condition embeddings to be progressively refined by emerging visual evidence.

Through systematic analysis of failure cases, we find that a considerable portion of failure cases do not stem from inadequate mask quality, but rather from *semantic misalignment*: the model *segments correctly but matches incorrectly*. As illustrated in Figure 2, in many failed predictions, at least one candidate mask already exhibits high overlap with the ground truth. This indicates a gap between language understanding and mask generation—language semantics are often used as static prompts or post-hoc matching signals, rather than actively participating in the iterative mask generation process.

We attribute this issue to the shallow and largely unidirectional semantic integration in existing pipelines. While LLMs provide strong condition embeddings, these representations are typically decoupled from the intermediate generation states (*e.g.*, evolving mask hypotheses or query representations), causing the generation process to be dominated by visual cues. As a result, models may produce visually plausible candidates that fail to satisfy fine-grained semantic constraints expressed in language, especially under ambiguous attributes or relational descriptions.

This observation suggests that language semantic grounding in LLM-conditioned segmentation should not be treated purely as a post-hoc selection problem. Instead, language semantics should participate directly in the generation dynamics, continuously shaping how intermediate representations evolve. Meanwhile, condition embeddings should remain adaptive, progressively absorbing visual evidence as candidate masks become more localized and structured. These considerations motivate a decoding principle where language and vision interact in a deep, continuous, and *bidirectional* manner throughout mask generation.

Motivated by this insight, we propose **FlowSeg**, which introduces *dynamic semantic guidance* via *bidirectional semantic flow* between intermediate decoding states and LLM-derived condition embeddings. FlowSeg allows language conditions to guide mask generation at each decoding stage, while enabling condition embeddings to be progressively updated by emerging visual evidence, yielding semantically grounded mask representations and visually aligned conditions for robust matching. A lightweight boundary-aware refinement module is further incorporated to selectively enhance uncertain boundary regions without perturbing confident interiors.

We evaluate FlowSeg on representative benchmarks that emphasize fine-grained semantic grounding, including referring expression segmentation (RefCOCO, RefCOCO+, RefCOCOg) (Kazemzadeh et al., 2014; Mao et al., 2016) and reasoning segmentation (ReasonSeg (Lai et al., 2024)). As shown in Figure 1, FlowSeg consistently outperforms recent state-of-the-art methods across all benchmarks, with more pronounced gains on challenging settings involving complex descriptions and implicit reasoning. Additional analysis on baseline (Wang et al., 2025) failure cases shows that FlowSeg recovers a large portion of errors associated with semantic misalignment, indicating that the improvements mainly come from better language–mask alignment rather than stronger mask generation alone.

In summary, this work makes the following contributions:

- We identify and empirically verify a previously underexplored failure mode in query-based LLM-conditioned segmentation, termed *semantic misalignment*, where accurate mask candidates are generated but incorrectly selected. Through oracle upper-bound analysis and systematic failure case recovery, we show

that a major performance bottleneck lies in candidate *selection* rather than candidate *generation*.

- We propose **FlowSeg**, which transforms static language prompting into *bidirectional semantic co-evolution*: LLM-derived condition embeddings continuously guide query refinement at each decoder layer, while being progressively grounded in emerging visual evidence. This bidirectional, layer-wise interaction is architecturally distinct from prior cross-modal attention where language representations remain fixed during decoding.

- We design a lightweight Boundary-Aware Refinement (BAR) module, motivated by the empirical observation that after BSF resolves global semantic misalignment, residual errors concentrate at object boundaries. BAR selectively corrects these uncertain regions without disturbing confident interiors, providing a hierarchical complement to BSF.

## 2. Related Work

**Language-Guided Segmentation.** Early referring segmentation methods (Kazemzadeh et al., 2014; Hu et al., 2016; Li et al., 2018; Yu et al., 2018; Ye et al., 2019) primarily relied on visual grounding techniques to localize objects described by natural language expressions. Recent advances leverage large language models (LLMs) to enhance semantic understanding. LISA (Lai et al., 2024) introduces the `<SEG>` token and proposes reasoning segmentation, enabling models to handle complex reasoning and world knowledge. Subsequent approaches (Xia et al., 2024) proposed more specialized tokens to handle diverse scenarios, such as addressing non-existent referents. SAM4MLLM (Chen et al., 2024) integrates Segment Anything Model (SAM) (Kirillov et al., 2023b) with multi-modal LLMs for pixel-aware tasks. More recently, several works have extended MLLMs with dedicated mask decoders for universal pixel-wise perception (Zhang et al., 2024b), leveraged LLM reasoning for universal image and video segmentation (Wei et al., 2024), or unified segmentation foundation models into a shared token space for dense grounded understanding (Yuan et al., 2025). Other frameworks further extend the foundation segmentation paradigm to handle category-specific and multi-mask scenarios (Wang et al., 2025). Despite these advances, in query-based mask decoding frameworks, language embeddings are still primarily used as static guidance or post-hoc matching signals, leaving semantic-visual interaction during mask generation underexplored.

**Transformer-based Segmentation.** DETR (Carion et al., 2020) revolutionizes object detection by formulating it as a direct set prediction problem using a transformer encoder-decoder architecture with learnable object queries, eliminating hand-crafted components like non-maximum suppression. Building on DETR's success, more studies (Zheng et al., 2021; Strudel et al., 2021; Xie et al., 2021; Zhang et al., 2021) focus on transformer-based segmentation, where MaskFormer (Cheng et al., 2021b) unifies panoptic, instance, and semantic segmentation through mask classification with transformer decoders. Mask2Former (Cheng et al., 2022) introduces masked attention that constrains cross-attention to foreground regions of predicted masks, achieving state-of-the-art performance across segmentation tasks. These query-based architectures generate fixed sets of mask proposals (typically 100 or 200 queries) through iterative refinement across decoder layers. However, the queries attend exclusively to visual features, with text conditions incorporated only post-hoc for classification, creating a semantic-visual decoupling during the generation process. While some prior works introduce cross-modal attention to inject language guidance into visual queries (Zhang et al., 2024b; Wei et al., 2024; Wang et al., 2025), these designs treat the language representation as a *fixed* key/value source throughout decoding: language informs vision, but visual evidence never reshapes the language embedding. Our work introduces a fundamentally different paradigm where language and vision engage in *bidirectional co-evolution* across decoder layers—condition embeddings actively guide query refinement while being simultaneously updated by emerging visual evidence, enabling tight semantic-visual alignment during mask generation rather than only at the final selection stage.

## 3. Method

### 3.1. Overall Architecture

FlowSeg targets **LLM-conditioned segmentation**: given an image $\mathbf{I}$ and a text instruction $\mathbf{T}$ (optionally containing phrase spans, e.g., `segment <p>the red car</p>`), the model outputs a pixel-wise mask $\mathbf{M}$ corresponding to the referred concept. As illustrated in Figure 3, we adopt a standard LLM–segmentor scaffold commonly used in recent systems (Lai et al., 2024; Xia et al., 2024). Our contribution is orthogonal to this scaffold and focuses on redesigning the decoder-side semantic–visual interaction via *Bidirectional Semantic Flow* (Sec. 3.3). **Dual Visual Encoders.** Following common practice in LLM-driven segmentation systems (Wei et al., 2024; Wang et al., 2025), we employ two complementary encoders to balance semantic understanding and spatial precision. A **Vanilla Encoder** extracts high-level semantic features for vision–language alignment, which are projected into the LLM embedding space via a lightweight projector $\phi_{\text{vis}}$. In parallel, a **Segmentation Encoder** extracts fine-grained pixel features $\mathbf{F}_{\text{pix}}$ for accurate localization, which are directly consumed by the segmentation decoder. This design decouples "*semantic comprehen-*

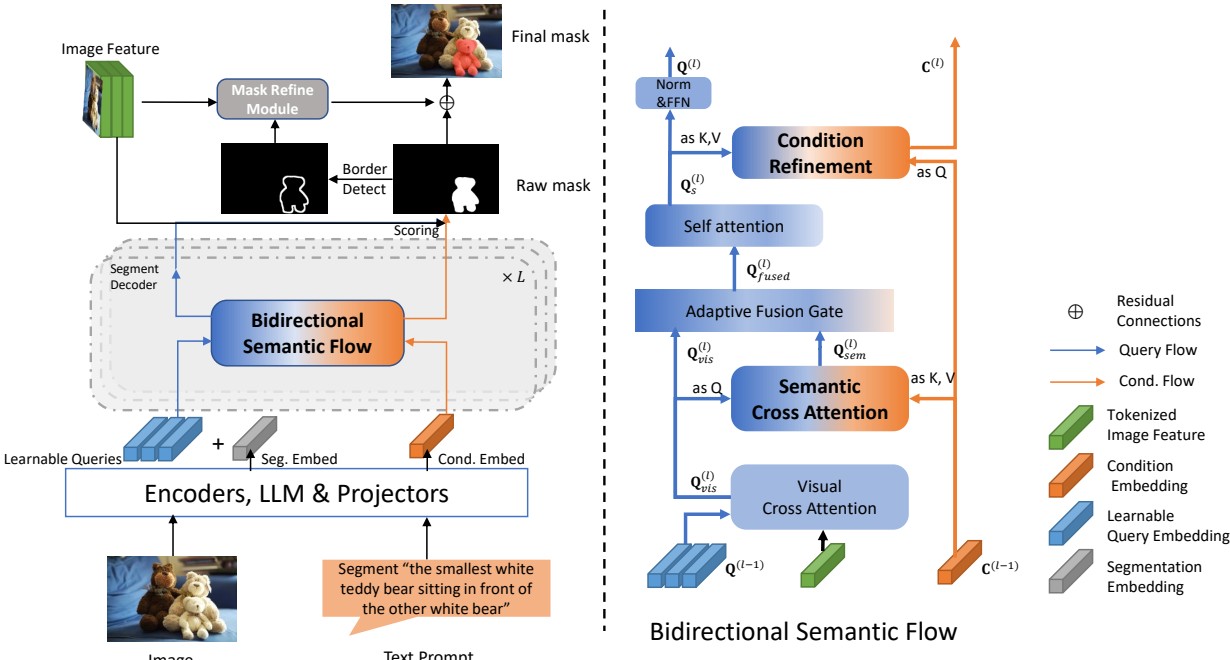

*Figure 3.* Main pipeline of proposed FlowSeg with Bidirectional Semantic Flow. Bidirectional Semantic Flow enables language condition embeddings to guide mask generation at each decoding stage, while progressively updating them with emerging query embeddings. A Boundary-Aware Mask Refinement further enhances boundary details without disrupting confident interior regions. (Best view in color.)

*sion*" from "*pixel-precise localization*", while still allowing them to interact through our decoder-level semantic flow.

**Language Model Processing.** We introduce special tokens <p>, </p>, and <SEG> following common practice in LLM-conditioned segmentation (Lai et al., 2024; Wang et al., 2025; Xia et al., 2024). The <p> and </p> tokens are used to explicitly mark the span of the referred phrase in the instruction, enabling phrase-level semantic extraction, while <SEG> indicates the segmentation output position. Let $\mathbf{H}_{\text{LLM}}$ denote the LLM hidden states. We extract (i) **condition embeddings** $\mathbf{C}_{\text{LLM}}$ from the hidden states corresponding to phrase spans, and (ii) **segmentation embeddings** $\mathbf{S}_{\text{LLM}}$ from the <SEG> token positions. These embeddings are projected to the decoder space via $\phi_{\text{llm}}$, yielding $\mathbf{C} = \phi_{\text{llm}}(\mathbf{C}_{\text{LLM}})$ and $\mathbf{S} = \phi_{\text{llm}}(\mathbf{S}_{\text{LLM}})$. The segmentation embedding $\mathbf{S}$ is added to the initial decoder queries, $\mathbf{Q}^{(0)} \leftarrow \mathbf{Q}^{(0)} + \mathbf{S}$, to provide global multi-modal context at the start of decoding.

**Segmentation Decoder.** FlowSeg employs a query-based segmentation decoder with $L$ stacked layers to iteratively generate masks. Given initial queries $\mathbf{Q}^{(0)}$, the decoder progressively refines them through cross-attention to pixel-level features and inter-query interactions, producing query representations $\mathbf{Q}_{\text{out}}$ and corresponding mask logits $\mathbf{M}_{\text{raw}}$. Unlike existing decoders, where language semantics are applied only at the final matching stage, our decoder integrates

*Bidirectional Semantic Flow* throughout the decoding process. Specifically, LLM-derived condition embeddings continuously interact with decoder queries during refinement, enabling semantic guidance to shape the decoding dynamics. The final mask selection is performed by matching $\mathbf{Q}_{\text{out}}$ with the refined condition embeddings $\mathbf{C}^L$. Optionally, mask predictions are further refined by a lightweight boundary-aware module (Sec. 3.4) to improve local precision. In all evaluated tasks (referring and reasoning segmentation), each forward pass processes a single referring expression, so condition embeddings from different expressions do not interact within the decoder.

### 3.2. Revisiting Baseline Architecture

Most LLM-conditioned segmentation methods employ a query-based decoder, where a fixed set of learnable queries is iteratively refined using visual features. Given image features $\mathbf{F}$ and initial queries $\mathbf{Q}^{(0)}$, the decoding follows: $\mathbf{Q}^{(l)} = \mathcal{D}(\mathbf{Q}^{(l-1)}, \mathbf{F})$, where $\mathcal{D}(\cdot)$ denotes a standard decoder layer with visual cross-attention and self-attention. An LLM encodes language semantics into condition embeddings $\mathbf{C}$ but is not involved in query refinement. Instead, semantic grounding is performed only after decoding, by matching final queries $\mathbf{Q}^{(L)}$ with $\mathbf{C}$, i.e., $s_i = \text{sim}(\mathbf{Q}_i^{(L)}, \mathbf{C})$. As a result, query evolution is driven largely by visual cues, while linguistic constraints are applied only at the final matching. This decoupled design leads to *se-*

*mantic misalignment*: accurate mask candidates may be generated during decoding, yet fail to be selected due to insufficient semantic guidance during refinement.

### 3.3. Bidirectional Semantic Flow

Condition embeddings extracted by the LLM are typically used only at the final matching stage, leaving the decoding dynamics largely vision-driven. This design gives rise to semantic misalignment, where visually accurate mask candidates may already exist but fail to be selected due to insufficient semantic guidance during refinement. To address this, we introduce **Bidirectional Semantic Flow**, a decoding paradigm in which language semantics actively participate in the decoding dynamics. Instead of serving as static matching cues, LLM-derived condition embeddings continuously interact with decoder queries throughout all decoder layers. Condition embeddings guide query refinement during mask generation, while being simultaneously refined by emerging visual evidence from the decoder. This bidirectional interaction enables language and vision to co-evolve during decoding, resulting in semantically grounded queries and more reliable mask–condition alignment.

**Semantic Cross-Attention.** To inject semantic guidance into query refinement, we introduce semantic cross-attention after visual cross-attention. This design allows each query to explicitly incorporate linguistic constraints during its iterative refinement, rather than relying solely on visual similarity. At decoder layer $l$, queries attend to condition embeddings as:

$$\mathbf{Q}_{\text{sem}}^{(l)} = \text{MultiHeadAttn}\Big(\mathbf{Q}_{\text{vis}}^{(l)}, \mathbf{C}^{(l-1)}, \mathbf{C}^{(l-1)}\Big), \quad (1)$$

where condition embeddings act as keys and values. This operation allows linguistic constraints to directly influence query updates during iterative decoding.

Importantly, semantic cross-attention does not replace visual reasoning. Instead, it complements visual cross-attention by constraining query updates with language-aware signals, ensuring that the refinement trajectory remains consistent with the given instruction. This mechanism directly addresses the semantic misalignment issue by allowing language semantics to shape mask generation throughout the decoding process, rather than only at the final matching stage.

**Adaptive Fusion Gates.** The importance of visual and semantic cues varies across queries and decoding stages. Initially, a balanced integration of visual features and semantic constraints is crucial for establishing coarse spatial-semantic hypotheses. As decoding progresses, however, the model increasingly relies on semantic guidance to refine mask-condition alignment and resolve ambiguities, ensuring consistency with the input instruction. To accommodate

this variability, we introduce adaptive fusion gates to dynamically balance their contributions:

$$\mathbf{g}^{(l)} = \sigma\Big(\mathbf{W}_g \cdot [\mathbf{Q}_{\text{vis}}^{(l)} \| \mathbf{Q}_{\text{sem}}^{(l)}]\Big), \quad (2)$$

where $\sigma(\cdot)$ denotes the sigmoid function and $\|$ denotes feature concatenation. The fused query representation is then obtained as:

$$\mathbf{Q}_{\text{fused}}^{(l)} = \mathbf{g}^{(l)} \odot \mathbf{Q}_{\text{vis}}^{(l)} + \Big(1 - \mathbf{g}^{(l)}\Big) \odot \mathbf{Q}_{\text{sem}}^{(l)}. \quad (3)$$

After adaptive fusion gates, detailed query interactions are retained through a standard **self-attention** mechanism. The fused queries $\mathbf{Q}_{\text{fused}}$ attend to each other to capture global context and dependencies among different queries, resulting in refined queries $\mathbf{Q}_{\text{s}}^{(l)}$:

$$\mathbf{Q}_{\text{s}}^{(l)} = \text{MultiHeadAttn}\Big(\mathbf{Q}_{\text{fused}}^{(l)}, \mathbf{Q}_{\text{fused}}^{(l)}, \mathbf{Q}_{\text{fused}}^{(l)}\Big). \quad (4)$$

This adaptive mechanism allows each query to modulate the degree of semantic influence according to its current refinement state. Rather than enforcing uniform constraints, the gate enables a smooth transition from balanced multimodal integration to semantically constrained refinement as decoding progresses.

**Condition Refinement.** Semantic cross-attention enables language conditions to guide query refinement. However, to establish a truly bidirectional interaction, semantic representations themselves should evolve by absorbing visual evidence during decoding. We therefore introduce *condition refinement*, which allows condition embeddings to be updated based on the current decoder state. At decoder layer $l$, condition embeddings attend to the refined queries via:

$$\mathbf{C}^{(l)} = \mathbf{C}^{(l-1)} + \text{MultiHeadAttn}\Big(\mathbf{C}^{(l-1)}, \mathbf{Q}_{\text{s}}^{(l)}, \mathbf{Q}_{\text{s}}^{(l)}\Big), \quad (5)$$

where $\mathbf{Q}_{\text{s}}^{(l)}$ denotes the self-attended queries. This update allows condition embeddings to progressively incorporate localized visual context as decoding proceeds.

By enabling condition embeddings to co-evolve with decoder queries, condition refinement closes the semantic–visual feedback loop. Together with semantic cross-attention, it forms a bidirectional semantic flow that continuously aligns language semantics with mask generation throughout decoding.

**Complete Forward Process.** We summarize the forward computation of a decoder layer with Bidirectional Semantic Flow in Algorithm 1. At each layer, decoder queries are first updated by visual cross-attention. Semantic guidance is then injected via semantic cross-attention and adaptively fused with visual features. The fused queries undergo self-attention to model inter-query dependencies, after which

**Algorithm 1** Bidirectional Semantic Flow Forward Process.

**Require:** Query features $\mathbf{Q}^{(l-1)}$, Condition embeddings $\mathbf{C}^{(l-1)}$, Image Features $\mathbf{F}$
**Ensure:** Updated Query $\mathbf{Q}^{(l)}$, Refined Condition $\mathbf{C}^{(l)}$
    **Visual Cross-Attention**:
1:  $\mathbf{Q}_{\text{vis}}^{(l)} \leftarrow \text{MultiHeadAttn}(\mathbf{Q}^{(l-1)}, \mathbf{F}, \mathbf{F})$
    **Semantic Cross-Attention with Adaptive Fusion**:
2:  $\mathbf{Q}_{\text{sem}}^{(l)} \leftarrow \text{MultiHeadAttn}(\mathbf{Q}_{\text{vis}}^{(l)}, \mathbf{C}^{(l-1)}, \mathbf{C}^{(l-1)})$
3:  $\mathbf{g}^{(l)} \leftarrow \sigma(\mathbf{W}_g \cdot [\mathbf{Q}_{\text{vis}}^{(l)} \| \mathbf{Q}_{\text{sem}}^{(l)}])$
4:  $\mathbf{Q}_{\text{fused}}^{(l)} \leftarrow \mathbf{g}^{(l)} \odot \mathbf{Q}_{\text{vis}}^{(l)} + (1 - \mathbf{g}^{(l)}) \odot \mathbf{Q}_{\text{sem}}^{(l)}$
    **Self-Attention**:
5:  $\mathbf{Q}_{\text{s}}^{(l)} \leftarrow \text{MultiHeadAttn}(\mathbf{Q}_{\text{fused}}^{(l)}, \mathbf{Q}_{\text{fused}}^{(l)}, \mathbf{Q}_{\text{fused}}^{(l)})$
    **Condition Refinement**:
6:  $\mathbf{C}^{(l)} \leftarrow \mathbf{C}^{(l-1)} + \text{MultiHeadAttn}(\mathbf{C}^{(l-1)}, \mathbf{Q}_{\text{s}}^{(l)}, \mathbf{Q}_{\text{s}}^{(l)})$
    **Feed-Forward Network**:
7:  $\mathbf{Q}^{(l)} \leftarrow \text{FFN}(\mathbf{Q}_{\text{s}}^{(l)})$
8:  **return** $\mathbf{Q}^{(l)}, \mathbf{C}^{(l)}$

condition embeddings are refined by attending to the updated queries. This process is repeated across decoder layers, enabling continuous and reciprocal semantic–visual interaction throughout decoding.

### 3.4. Boundary-Aware Mask Refinement

While Bidirectional Semantic Flow resolves global semantic misalignment, we observe that residual errors are often concentrated around object boundaries. To complement semantic alignment with local precision, we introduce a lightweight boundary-aware refinement module that selectively improves uncertain boundary regions without affecting confident interiors.

**Boundary Localization.** Given raw mask probability map $\mathbf{M}_{\text{prob}}$, boundary pixels are identified using a morphological gradient operation (Rivest et al., 1993; Dougherty, 1992):

$$\mathbf{B} = \mathbb{I}\big[(\text{dilate}(\mathbf{M}_{\text{prob}}) - \text{erode}(\mathbf{M}_{\text{prob}})) > \epsilon\big], \quad (6)$$

where $\mathbf{B}$ denotes the boundary mask and $\epsilon$ controls boundary sensitivity.

**Selective Refinement.** Refinement is applied as a bounded residual update restricted to $\mathbf{B}$:

$$\Delta\mathbf{M} = \tanh\big(f_{\text{refine}}([\mathbf{M}_{\text{raw}} \| f_{\text{comp}}(\mathbf{F}_{\text{pix}})])\big) \cdot \alpha,$$
$$\mathbf{M}_{\text{refined}} = \mathbf{M}_{\text{raw}} + \Delta\mathbf{M} \odot \mathbf{B}, \quad (7)$$

where $f_{\text{refine}}$ is a light-weight refinement network, $f_{\text{comp}}$ compresses pixel features and $\alpha$ is a learnable scale. This design follows an *enhancement-not-replacement* principle, ensuring that refinements are confined to ambiguous boundaries. Overall, this module provides a simple and efficient complement to Bidirectional Semantic Flow, improving

boundary accuracy without interfering with decoder-side semantic reasoning.

### 3.5. Training Objectives

FlowSeg is trained end-to-end with a multi-task objective that jointly optimizes language modeling and segmentation:

$$\mathcal{L}_{\text{total}} = \mathcal{L}_{\text{llm}} + \mathcal{L}_{\text{seg}}. \quad (8)$$

**Language Modeling Loss.** We adopt the standard next-token prediction loss to preserve the instruction-following and reasoning capability of the LLM:

$$\mathcal{L}_{\text{llm}} = -\sum_{t=1}^{L} \log P(y_t \mid y_{<t}, \mathbf{I}, \mathbf{T}), \quad (9)$$

where $\mathbf{I}$ and $\mathbf{T}$ denote the input image and text instruction, respectively.

**Segmentation Loss.** The segmentation objective supervises both mask quality and semantic alignment. Following query-based segmentation practice, we apply Hungarian matching between predicted masks and ground-truth annotations, and minimize:

$$\mathcal{L}_{\text{seg}} = \mathcal{L}_{\text{CE}} + \lambda_{\text{dice}}\mathcal{L}_{\text{dice}} + \lambda_{\text{mask}}\mathcal{L}_{\text{mask}}, \quad (10)$$

where $\mathcal{L}_{\text{CE}}$ enforces correct semantic categorization of queries, while $\mathcal{L}_{\text{dice}}$ and $\mathcal{L}_{\text{mask}}$ supervise pixel-wise mask accuracy. To stabilize optimization and facilitate semantic propagation across layers, deep supervision is applied at all decoder stages.

## 4. Experiments

### 4.1. Experimental Setup

**Datasets.** We conduct extensive experiments on multiple segmentation benchmarks to evaluate FlowSeg: **Referring Expression Segmentation**: We use RefCOCO (Kazemzadeh et al., 2014), Ref-COCO+ (Kazemzadeh et al., 2014), and RefCOCOg (Mao et al., 2016), which contain 142K, 142K, and 85K referring expressions respectively. **Reasoning Segmentation**: We evaluate on ReasonSeg (Lai et al., 2024), a challenging dataset requiring complex reasoning to segment objects based on implicit descriptions.

**Implementation Details.** FlowSeg is built upon Qwen-3 (Yang et al., 2025) as the language model, SigLIP2-so400m (Zhai et al., 2023) as the vanilla encoder, and SAM-ViT-Large (Kirillov et al., 2023a) as the segmentation encoder. The decoder follows Mask2Former (Cheng et al., 2022) architecture with $N = 200$ learnable queries.

We adopt a three-stage training paradigm: (1) segmentor pre-training for 36 epochs, (2) vision-language alignment

for 1 epoch, and (3) multi-task joint training for 2 epochs. During stage 3, we use AdamW optimizer with learning rate $4 \times 10^{-5}$, weight decay 0.05, and batch size 8 per GPU across 8 GPUs. We set $\epsilon = 0.1$ for morphological boundary detection, loss weights as $\lambda_{cls} = 2.0$, $\lambda_{mask} = 5.0$, $\lambda_{dice} = 5.0$ following standard practice (Cheng et al., 2022). All experiments are conducted on NVIDIA H20 GPUs. The proposed BSF module introduces only $+5.93$M parameters ($+0.12\%$) and $+4.28$ ms per-sample latency ($+1.39\%$) compared to the baseline, with negligible computational overhead. Further implementation details and a complete overhead breakdown are provided in the appendix A.2.

**Evaluation Metrics.** For referring and reasoning segmentation, we report cumulative Intersection over Union (cIoU) and global mean Intersection over Union (gIoU) following prior works (Lai et al., 2024). Following LISA (Lai et al., 2024) and X-SAM (Wang et al., 2025), we adopt an expression-level evaluation protocol where each forward pass receives a single referring expression.

## 4.2. Main Results

**Referring Expression Segmentation.** Table 1 presents the results on RefCOCO, RefCOCO+, and RefCOCOg validation and test sets. FlowSeg consistently outperforms all comparison methods across all dataset splits. Compared to the prior state-of-the-art work X-SAM (Wang et al., 2025), FlowSeg achieves average absolute improvements of 0.8%, 2.7%, and 2.4% cIoU on RefCOCO, RefCOCO+, and RefCOCOg respectively. Notably, the performance improvement is more pronounced on the challenging RefCOCO+ and RefCOCOg benchmark. These results are consistent with the goal of bidirectional semantic flow: improving query-condition alignment during mask decoding. Additionally, the boundary-aware mask refinement module further contributes to handling intricate object details in complex scenarios, boosting the overall quality. Furthermore, FlowSeg consistently surpasses recent MLLM-based methods, including SAM4MLLM-7B (Chen et al., 2024), PSALM (Zhang et al., 2024b), HyperSeg (Wei et al., 2024), Sa2VA-8B (Yuan et al., 2025), X-SAM (Wang et al., 2025), reinforcing the effectiveness of explicit semantic interaction between queries and language conditions.

**Reasoning Segmentation.** Table 2 shows the results on ReasonSeg benchmark. FlowSeg achieves 54.7% cIoU on ReasonSeg test set, outperforming X-SAM by 13.7% and surpassing prior methods including LISA (Lai et al., 2024) and HyperSeg (Wei et al., 2024). It is worth noting that the ReasonSeg validation set is a small split with only 340 cases, which may lead to unstable evaluation; we therefore report both splits for completeness and encourage readers to refer primarily to the test split results (Wang et al., 2025). The significant improvement indicates that our dynamic condition

refinement mechanism enables better semantic understanding of complex reasoning descriptions. The boundary-aware mask refinement further enhances segmentation quality by incorporating fine-grained visual details.

## 4.3. Ablation Studies

We conduct comprehensive ablation studies to validate the effectiveness of each proposed component. All ablations are performed on RefCOCO/+/g validation set with cIoU unless otherwise specified.

**Component Contribution Analysis.** Table 3 analyzes the incremental impact of each proposed component on RefCOCO, RefCOCO+, and RefCOCOg validation sets. Starting from the baseline (row 1), we first introduce **Semantic Refinement (SR)** (*i.e.*, Semantic Cross-Attention with Adaptive Fusion) and **Condition Refinement (CR)**, which constitute our proposed **Bidirectional Semantic Flow**. Adding SR (row 2) enables direct query guidance from text, while further incorporating CR (row 3) establishes the reverse feedback loop, significantly improving semantic alignment. Finally, we integrate the **Boundary-Aware mask Refinement (BAR)** module (row 4) to enhance boundary precision. Each component contributes to consistent performance gains across all benchmarks, with the full configuration achieving 84.2% average cIoU, validating the effectiveness of both deep semantic interaction and fine-grained boundary refinement. We provide a detailed discussion on the impact of the Boundary-Aware Mask Refinement on boundary quality in the following subsection.

**Backbone-Controlled Ablation.** To verify that the performance gains of FlowSeg stem from the proposed Bidirectional Semantic Flow architecture rather than the choice of language model backbone, we evaluate FlowSeg with Phi-3-3.8B—the same backbone used by X-SAM (Wang et al., 2025)—under identical training settings. As shown in Table 4, simply upgrading the LLM backbone (X-SAM w/ Qwen3) yields only marginal improvements, whereas FlowSeg with the same Phi-3 backbone as X-SAM consistently outperforms it across all benchmarks. These results confirm that the performance improvements are attributable to the architectural contribution of Bidirectional Semantic Flow.

**Analysis of Misalignment Cases.** Our core motivation holds that in query-based segmentation, correct mask candidates are often already generated but incorrectly selected. To validate this, we first compute the oracle cIoU upper bound by selecting the best-matching candidate from the full query set for each sample:

The oracle upper bounds for both methods are nearly identical ($\sim$91%), indicating that both models generate comparably high-quality candidates. The performance gap therefore

*Table 1.* Referring expression segmentation results on RefCOCO, RefCOCO+, and RefCOCOg.

| Method | RefCOCO | | | RefCOCO+ | | | RefCOCOg | |
|---|---|---|---|---|---|---|---|---|
| | val | testA | testB | val | testA | testB | val | test |
| LISA-7B (Lai et al., 2024) | 74.9 | 79.1 | 72.3 | 65.1 | 70.8 | 58.1 | 67.9 | 70.6 |
| PixelLM-7B (Ren et al., 2024) | 73.0 | 76.5 | 68.2 | 66.3 | 71.7 | 58.3 | 69.3 | 70.5 |
| GSVA-7B (Xia et al., 2024) | 76.4 | 77.4 | 72.8 | 64.5 | 67.7 | 58.6 | 71.1 | 72.0 |
| SAM4MLLM-7B (Chen et al., 2024) | 79.6 | 82.8 | 76.1 | 73.5 | 77.8 | 65.8 | 74.5 | 74.6 |
| PSALM (Zhang et al., 2024b) | 83.6 | 84.7 | 81.6 | 72.9 | 75.5 | 70.1 | 73.8 | 74.4 |
| HyperSeg (Wei et al., 2024) | 84.8 | 85.7 | 83.4 | 79.0 | 83.5 | 75.2 | 79.4 | 78.9 |
| Sa2VA-8B (Yuan et al., 2025) | 81.6 | - | - | 76.2 | - | - | 78.7 | - |
| X-SAM (Wang et al., 2025) | 85.1 | 87.1 | 83.4 | 78.0 | 81.0 | 74.4 | 83.8 | 83.9 |
| FlowSeg (Ours) | **85.8** | **87.4** | **84.8** | **80.2** | **84.5** | **76.9** | **86.5** | **86.1** |

*Table 2.* Reasoning segmentation results on ReasonSeg validation and test sets. We report gIoU and cIoU. ft: finetuned on reaseg dataset.

| Method | Val | | Test | |
|---|---|---|---|---|
| | gIoU | cIoU | gIoU | cIoU |
| LISA-7B (Lai et al., 2024) | 44.0 | 46.0 | 36.8 | 34.1 |
| LISA-7B(ft.) (Lai et al., 2024) | 52.9 | 54.0 | 47.3 | 48.4 |
| HyperSeg (Wei et al., 2024) | 59.2 | **56.7** | - | - |
| X-SAM (Wang et al., 2025) | 56.6 | 32.9 | 57.8 | 41.0 |
| FlowSeg (Ours) | **62.4** | 49.2 | **60.5** | **54.7** |

*Table 3.* Ablation study on architectural components. SR: Semantic Refinement (Semantic Cross-Attention with Adaptive Fusion); CR: Condition Refinement; BAR: Boundary-Aware mask Refinement. Results are reported on RefCOCO/+/g val sets.

| Row | SR | CR | BAR | RefCOCO | RefCOCO+ | RefCOCOg | Avg. |
|---|---|---|---|---|---|---|---|
| 1 | | | | 85.0 | 78.3 | 84.1 | 82.4 |
| 2 | ✓ | | | 85.4 | 79.0 | 84.3 | 82.9 |
| 3 | ✓ | ✓ | | 85.6 | 79.9 | 86.2 | 83.9 |
| 4 | ✓ | ✓ | ✓ | **85.8** | **80.2** | **86.5** | **84.2 (+1.8)** |

*Table 4.* Backbone-controlled ablation on RefCOCO/+/g val sets (cIoU). * denotes results reproduced using official pretrained weights.

| Method | LLM | RefCOCO | RefCOCO+ | RefCOCOg |
|---|---|---|---|---|
| X-SAM* (Wang et al., 2025) | Phi-3-3.8B | 84.3 | 78.2 | 82.5 |
| X-SAM* (w/ Qwen3) | Qwen3-4B | 85.0 | 78.3 | 84.1 |
| FlowSeg (w/ Phi-3) | Phi-3-3.8B | 85.5 | 79.6 | 85.9 |
| FlowSeg (Ours) | Qwen3-4B | **85.8** | **80.2** | **86.5** |

*Table 5.* Oracle cIoU upper bound on RefCOCO/+/g val sets. Both models generate similarly high-quality candidates ( 91%); the performance gap reflects selection quality, not generation capacity.

| Method | RefCOCO | RefCOCO+ | RefCOCOg |
|---|---|---|---|
| X-SAM (Wang et al., 2025) | 91.31 | 91.30 | 91.16 |
| FlowSeg (Ours) | 91.47 | 91.44 | 91.38 |

mainly reflects *better candidate selection*, supporting that Bidirectional Semantic Flow improves semantic alignment at the selection stage. We further quantify this by evaluating FlowSeg on specific X-SAM failure cases (cIoU < 0.5 or 0.2) across three benchmarks, as shown in Table 6. FlowSeg achieves significantly higher average IoU on these subsets, rescuing 44.6% of the cIoU < 0.5 failed cases, further supporting the effectiveness of bidirectional flow for mitigating semantic misalignment.

**Effectiveness of Boundary-Aware Mask Refinement.** As BAR is specifically designed to refine uncertain boundary regions while leaving confident interiors intact, its primary effect is captured by the Boundary IoU (BIoU) metric rather than cIoU—this is by design, as the localized boundary correction is not well reflected in region-level overlap. We evaluate the impact of BAR using the Boundary IoU metric (Cheng et al., 2021a), computing both the cumulative

Boundary IoU (cBIoU) and the global mean Boundary IoU (gBIoU) on RefCOCO, RefCOCO+, and RefCOCOg validation sets. Results are reported in Table 7. The model with mask refinement achieves consistently higher Boundary IoU scores across all splits, demonstrating its capability to produce more precise segmentation boundaries. This improvement underscores the positive role of our proposed refinement in generating high-quality masks aligned with object boundaries.

### 4.4. Qualitative Analysis

We present qualitative comparisons between X-SAM and FlowSeg on referring expression segmentation task. Figure 4 illustrates the results on RefCOCO/+/g datasets. FlowSeg produces more accurate segmentation masks with clearer boundaries, especially for objects with ambiguous descriptions. More visualization results will be discussed in the appendix Sec. A.5.

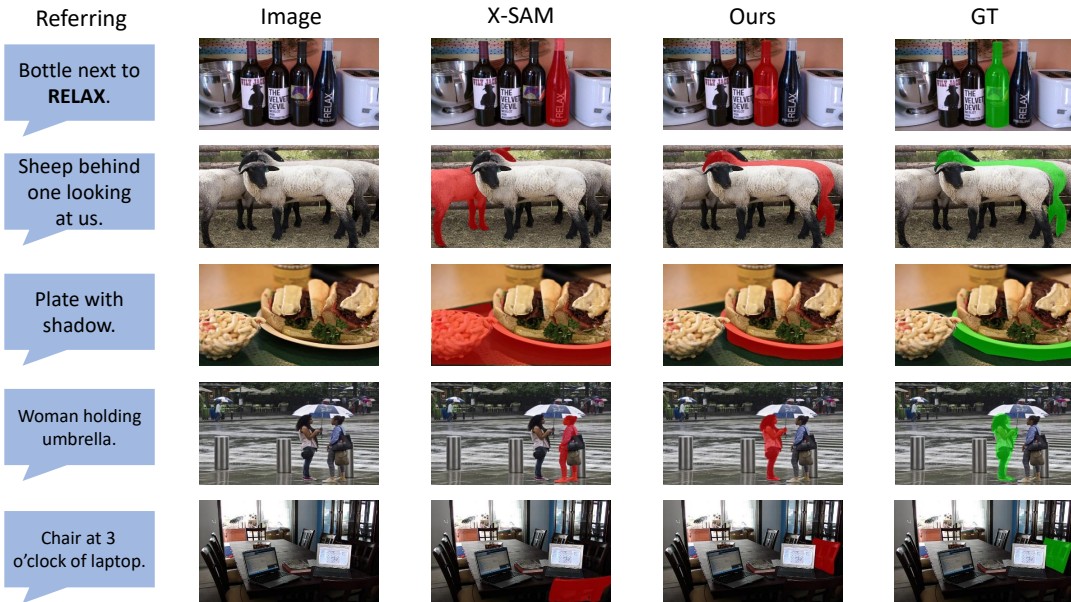

*Figure 4.* Qualitative comparison on RefCOCO/+/g. FlowSeg produces more accurate masks with finer details compared to X-SAM.

*Table 6.* Comparative performance on prior work (Wang et al., 2025) failure cases. The cIoU is reported for these specific subsets.

| Method | RefCOCO | RefCOCO+ | RefCOCOg | Avg. |
|---|---|---|---|---|
| Baseline (cIoU<0.5) | 2.7 | 4.7 | 6.3 | 4.6 |
| FlowSeg (Ours) | **42.9** | **46.1** | **51.7** | **49.2** |
| Improvement (Δ) | +40.2 | +41.4 | +45.4 | +44.6 |
| Baseline (cIoU<0.2) | 1.0 | 1.1 | 1.1 | 1.1 |
| FlowSeg (Ours) | **41.6** | **42.1** | **49.7** | **44.4** |
| Improvement (Δ) | +40.6 | +41.0 | +48.6 | +43.4 |

*Table 7.* Ablation study on the Boundary-Aware Mask Refinement module. We report cumulative Boundary IoU (cBIoU) and the global mean Boundary IoU (gBIoU) on RefCOCO, RefCOCO+, and RefCOCOg val sets.

| Setting | RefCOCO | | RefCOCO+ | | RefCOCOg | |
|---|---|---|---|---|---|---|
| | gBIoU | cBIoU | gBIoU | cBIoU | gBIoU | cBIoU |
| w/o Mask Refine | 70.6 | 67.8 | 67.4 | 63.0 | 72.0 | 68.7 |
| w/ Mask Refine | **72.2** | **70.0** | **68.1** | **64.1** | **73.5** | **70.4** |

## 5. Conclusion

In this work, we addressed the issue of *semantic misalignment* in query-based LLM-conditioned segmentation systems, where decoupling language semantics from iterative mask generation often leads to incorrect mask selection despite high-quality candidates. We proposed **FlowSeg**, a framework that introduces *Bidirectional Semantic Flow* to enable deep, reciprocal interaction between language condition embeddings and intermediate decoding states. By incorporating semantic cross-attention, adaptive fusion gates, and condition refinement, FlowSeg ensures that linguistic constraints actively guide mask evolution while condition embeddings progressively absorb visual evidence. Additionally, a lightweight boundary-aware refinement module was introduced to enhance local mask precision. Extensive experiments across RefCOCO, RefCOCO+, RefCOCOg, and ReasonSeg benchmarks demonstrate that FlowSeg consistently achieves state-of-the-art performance, effectively resolving semantic grounding failures and producing high-quality segmentation masks.

## Impact Statement

Based on our novel perspective that the shallow participation of language semantics during iterative decoding is the root cause of semantic misalignment, we propose FlowSeg, a method based on bidirectional semantic flow, to address the LLM-conditioned segmentation problem. This approach can also be applied in other areas such as multi-modal content generation and interactive image editing, since the challenge of aligning high-level semantic constraints with fine-grained spatial outputs is widespread across various vision-language tasks. The limitation of this work is its reliance on explicit phrase markers for precise condition extraction, which may be less practical in unconstrained, purely conversational scenarios. Nonetheless, our method is highly adaptable and can be further optimized to support unformatted natural language instructions in the future.

**Acknowledgment.** This work was supported by the National Natural Science Foundation of China (Grant Nos. 62472033, U23A20314, and 92470203) and the Beijing Natural Science Foundation (Grant No. L242022).

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

# A. Technical Appendices and Supplementary Material

In the Appendix, we first provide more details on the Boundary-Aware Mask Refinement module, which is a key component of FlowSeg for improving mask boundary quality. Then, we present additional model and training details, followed by more experimental results on various benchmarks to demonstrate the effectiveness of our approach. Finally, we include visualization results for referring segmentation and reasoning segmentation tasks.

## A.1. More Method Details

**Boundary-Aware Mask Refinement Module.** The Boundary-Aware Mask Refinement module is designed to improve the boundary quality of predicted masks via lightweight networks. This module operates as a post-processing step after the decoder output, following an "enhancement-not-replacement" design philosophy.

**Architecture.** The refiner consists of three components. *Boundary Detection* applies morphological operations implemented via $3 \times 3$ max/min pooling, to identify boundary regions. A *Feature Compression Layer* $f_{comp}$ uses a $1 \times 1$ convolution to compress pixel features from 256 to $d = 16$ dimensions, reducing computational cost while preserving boundary information. A *Refinement Network* $f_{refine}$ is a lightweight CNN with two convolutional layers that takes concatenated mask logits, compressed features as input and outputs the refinement boundary map.

**Enhancement-Not-Replacement Design.** The refiner selectively modifies predictions only in boundary regions (where mask probability is uncertain) via residual correction, while high-confidence regions (clearly inside or outside) remain untouched as the boundary map is zero. This design ensures robustness even if the refiner module fails.

**Sensitivity Analysis of Boundary Threshold $\varepsilon$.** Table 8 reports BIoU performance across a range of $\varepsilon$ values on all three benchmarks. Performance is stable across $\varepsilon \in [0.05, 0.15]$, with $\varepsilon = 0.1$ achieving the best results in our evaluation. This suggests that the default choice is robust to exact hyperparameter tuning.

*Table 8.* Sensitivity analysis of boundary threshold $\varepsilon$ on RefCOCO, RefCOCO+, and RefCOCOg val sets (gBIoU / cBIoU). $\varepsilon = 1.0$ corresponds to no BAR (baseline).

| $\varepsilon$ | RefCOCO | | RefCOCO+ | | RefCOCOg | |
|---|---|---|---|---|---|---|
| | gBIoU | cBIoU | gBIoU | cBIoU | gBIoU | cBIoU |
| 1.0 (no BAR) | 70.6 | 67.8 | 67.4 | 63.0 | 72.0 | 68.7 |
| 0.05 | 71.7 | 69.6 | 67.9 | 63.8 | 73.2 | 69.9 |
| **0.1 (selected)** | **72.2** | **70.0** | **68.1** | **64.1** | **73.5** | **70.4** |
| 0.15 | 71.9 | 69.4 | 68.1 | 63.7 | 73.1 | 70.3 |
| 0.2 | 71.4 | 68.9 | 67.8 | 63.6 | 72.8 | 69.6 |

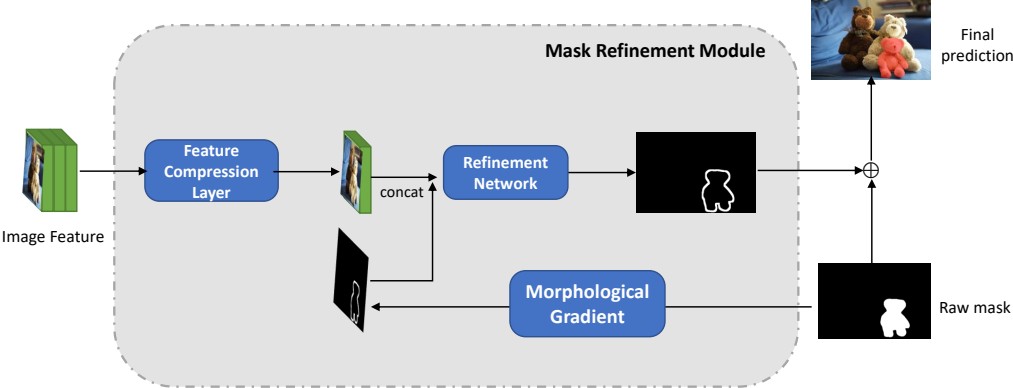

*Figure 5.* Detailed architecture of boundary-aware mask refinement module.

## A.2. More Model Details

**Model Framework.** FlowSeg employs a dual-encoder architecture, comprising a vanilla encoder (SigLIP2-so400m (Tschannen et al., 2025)) for semantic feature extraction and a segmentation encoder (SAM-L (Kirillov et al., 2023b)) for fine-grained pixel-level features. The **Bidirectional Semantic Flow** decoder incorporates semantic cross-attention and adaptive fusion gates, enabling continuous interaction between visual queries and LLM-extracted condition embeddings (Qwen-3-4B (Yang et al., 2025)). The **Boundary-Aware Mask Refinement** module is placed after decoder output, operating on 1/4-scale mask logits with image pixel features.

**Projector Architecture.** The projector $\phi_{\text{llm}}$ that maps LLM hidden states to the decoder space is a two-layer MLP:

$$\mathbf{f}_{\text{seg}} = \text{Linear}_2(\text{GELU}(\text{Linear}_1(\mathbf{h}_{\text{LLM}}))), \tag{11}$$

where $\mathbf{h}_{\text{LLM}} \in \mathbb{R}^{3072}$ is the LLM hidden state (extracted at <p> phrase-token positions, with $D_{\text{LLM}} = 3072$ for Qwen3-4B), and $\mathbf{f}_{\text{seg}} \in \mathbb{R}^{256}$ is the projected condition embedding fed into the segmentation decoder ($D_{\text{seg}} = 256$ for the Mask2Former-style decoder).

**Computational Overhead.** Table 9 reports the parameter count and end-to-end inference overhead of FlowSeg relative to the Qwen3 baseline, measured on RefCOCO val ($N = 10268$ samples).

*Table 9.* Computational overhead of BSF compared to the Qwen3 baseline.

| | Baseline (Qwen3) | FlowSeg |
|---|---|---|
| Segmentor params | 328.9M | 334.8M (+5.93M) |
| **Total params** | **4816.7M** | **4822.6M (+0.12%)** |
| Inference memory | 17.848G | 17.860G (+0.012G) |
| Latency (ms/sample) | 307.20 | 311.48 (+4.28 ms, +1.39%) |
| GFLOPs | 18525.48 | 18543.00 (+17.52, +0.09%) |

Although the enhanced decoder layer adds $\sim$14.81% FLOPs *per layer*, the segmentation decoder is a small fraction of the full pipeline dominated by the MLLM and SAM encoder, resulting in near-zero end-to-end overhead.

## A.3. More Training Details

**Multi-stage Training Strategy.** Model training proceeds in three stages (Wang et al., 2025). *Stage 1* fine-tunes the segment encoder and decoder on COCO Panoptic (118K samples) for 36 epochs with batch size 64. *Stage 2* trains dual projectors on LLaVA conversation data for 1 epoch with batch size 256. *Stage 3* jointly fine-tunes the entire model on mixed datasets for 2 epoch with equivalent batch size 64. Training strategies and datasets are following X-SAM (Wang et al., 2025).

**Stage 3 Component Training.** Both the Bidirectional Semantic Flow decoder and the Boundary-Aware Refiner are introduced in Stage 3 and trained end-to-end. The semantic cross-attention, adaptive fusion gates, and condition refinement modules use the base learning rate of 4e-5. The Boundary-Aware Mask Refinement module is trained with the same learning rate.

## A.4. More Experimental Results

**Generic Segmentation.** Although our method primarily focuses on LLM-Conditioned Segmentation, aiming to leverage the rich semantic information extracted by LLMs to guide segmentation, we also provide results on generic segmentation tasks to demonstrate the robustness of our approach. We validate FlowSeg on semantic, instance, and panoptic segmentation, comparing with prior methods.

Table 10 presents the results of generic segmentation on the COCO-Panoptic (Lin et al., 2014) dataset. FlowSeg achieves state-of-the-art performance compared to the previous works, suggesting that the Boundary-Aware Refinement module improves mask quality without sacrificing overall segmentation performance. Notably, its performance even approaches that of methods specifically trained for close-set segmentation. This multi-object setting also suggests that BSF does not introduce obvious semantic contamination across simultaneously decoded entities in practice.

**Semantic vs. Visual Cross-Attention.** To isolate whether performance gains stem from *semantic conditioning* or simply

*Table 10.* Comparison of Generic Segmentation. We compare different methods on COCO-Panoptic benchmark. Underlined for the second best.

| Method | Encoder | PQ | AP | mIoU |
|---|---|---|---|---|
| Mask2Former(Cheng et al., 2022) | Swin-L | 57.8 | 48.6 | 67.4 |
| X-Decoder(Zou et al., 2023a) | DaViT-B | 56.2 | 45.8 | 66.0 |
| SEEM(Zou et al., 2023b) | DaViT-B | 56.1 | 46.4 | 66.3 |
| OMG-LLaVA(Zhang et al., 2024a) | ConvNeXt-XXL | 53.8 | - | - |
| PSALM(Zhang et al., 2024b) | Swin-B | 55.9 | 45.7 | 66.6 |
| X-SAM (Wang et al., 2025) | SAM-L | 54.7 | 47.0 | 66.5 |
| FlowSeg (Ours) | SAM-L | **56.1** | **47.2** | **67.4** |

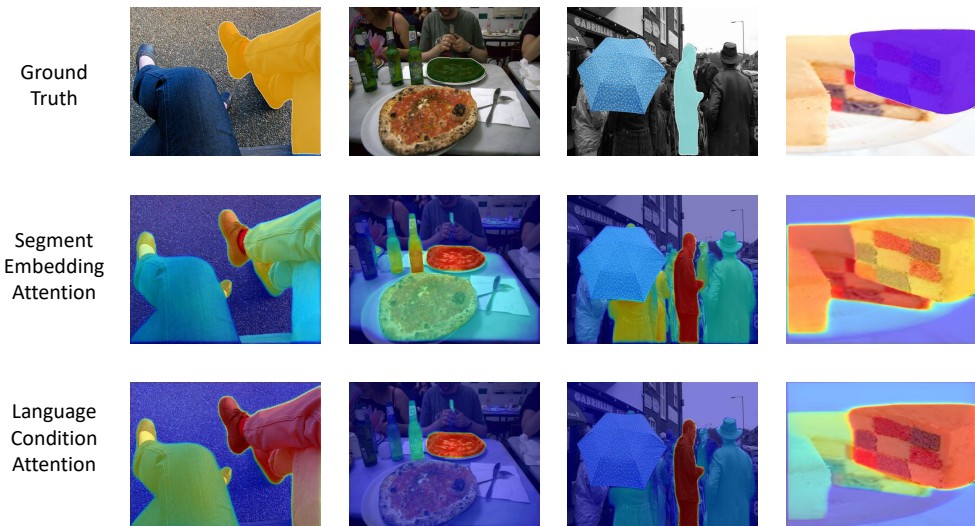

*Figure 6.* Visualization of attention weights in the Bidirectional Semantic Flow, where darker red indicates higher attention. It is observed that the segment embedding attention is distributed across various potential candidate objects with commonalities, while the language condition attention is precisely localized on the specific candidate referred to by the instruction.

from adding more attention operations, we construct an ablation decoder where all conditional embeddings are replaced with pixel-decoder visual features while keeping the architecture depth and width identical (same parameter count and attention operations). As shown in Table 11, this visual counterpart yields only marginal improvements over the baseline, while Semantic Cross-Attention robustly outperforms it across all benchmarks. This indicates that the gains are attributable to the semantic alignment mechanism, not increased model depth.

*Table 11.* Ablation: Semantic vs. Visual Cross-Attention. Both variants match in architecture depth and parameter count.

| Method | RefCOCO | RefCOCO+ | RefCOCOg |
|---|---|---|---|
| Baseline (default decoder) | 85.0 | 78.3 | 84.1 |
| + Visual Cross-Attn (equal-param ablation) | 85.1 | 78.5 | 84.3 |
| **+ Semantic Cross-Attn (FlowSeg)** | **85.8** | **80.2** | **86.5** |

**Stability of Condition Refinement.** To empirically verify that the bidirectional update mechanism does not cause semantic drift in early, uncertain decoding stages, we track the cosine similarity between the initial condition embedding $\mathbf{C}^{(0)}$ and the layer-$l$ output $\mathbf{C}^{(l)}$, averaged across RefCOCO/+/g validation sets. Results are shown in Table 12.

Early layers ($l \leq 3$) maintain high similarity ($> 0.86$): when visual queries are uncertain and diffuse, the cross-attention effectively acts as a soft information filter that suppresses unreliable visual noise. Substantial refinement occurs only in deeper layers ($l \geq 7$) once queries have matured. The similarity never collapses to zero or becomes negative, indicating that

*Table 12.* Layer-wise cosine similarity between $\mathbf{C}^{(0)}$ and $\mathbf{C}^{(l)}$.

| Layer $l$ | 1 | 2 | 3 | 4 | 5 | 6 | 7 | 8 | 9 |
|---|---|---|---|---|---|---|---|---|---|
| Cosine Sim. | 0.95 | 0.91 | 0.86 | 0.82 | 0.77 | 0.71 | 0.64 | 0.55 | 0.53 |

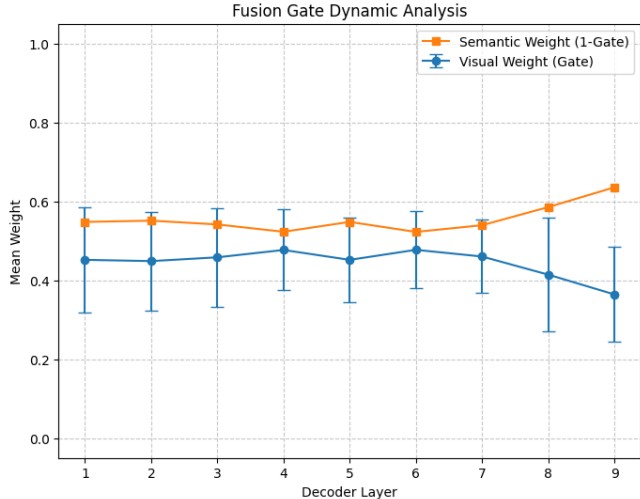

*Figure 7.* Distribution of fusion gate values across decoding process.

the residual connection anchors $\mathbf{C}^{(0)}$ as a persistent semantic reference throughout decoding.

### A.5. More Visualization Results

**Attention Weight Visualization.** To understand how the bidirectional semantic flow works, we visualize the attention weights in Figure 6. By examining the attention maps, we observe that the segment embedding attention effectively attends to various potential candidate objects in the scene that share commonalities, such as bottles, persons, or cakes. In contrast, the language condition attention is precisely localized on the specific candidate referred to by the condition instruction. The synergy between these two mechanisms ensures that the model not only identifies all plausible candidates but also accurately selects the target described by the user. This combined focus forms the core of our method's effectiveness in resolving complex multi-modal references.

**Fusion Gate Analysis.** Figure 7 shows the distribution of fusion gate values across different decoder layers. In early layers, the gate values are more balanced (around 0.5), indicating equal weighting of visual and semantic features. In deeper layers, the gate gradually biases towards semantic features (lower values), suggesting that the model increasingly relies on semantic guidance to refine the alignment and ensure the segmented regions stay consistent with the input prompt throughout the decoding process.

**Referring Segmentation.** Figure 8 shows the visualization results of FlowSeg in referring segmentation tasks. The results demonstrate that our Boundary-Aware Refinement module produces masks with cleaner and more accurate boundaries, especially for objects with complex shapes or fine details.

**Reasoning Segmentation.** Figure 9 shows the visualization results on the reasoning segmentation benchmark. FlowSeg effectively understands complex questions and generates masks with improved boundary quality. The refinement module particularly helps in cases where the reasoning requires precise object boundaries.

### A.6. Further Discussion of Limitations and Future Work

FlowSeg is evaluated on single-referent segmentation tasks (referring expression and reasoning segmentation), where each forward pass processes one text condition at a time. In this setting, different conditions are naturally decoupled in the batch dimension and no cross-expression semantic interference arises. We further validate robustness in multi-object panoptic

settings via the COCO-Panoptic benchmark (Table 10), where FlowSeg outperforms comparable generalist models without observable degradation.

Nevertheless, in denser simultaneous multi-entity decoding scenarios where multiple condition embeddings co-exist in a single decoder pass, cross-query semantic interactions could theoretically risk semantic contamination. A promising future direction is to introduce group-wise attention masking within the self-attention stage, constraining query interactions to within the same semantic entity group and further improving semantic independence.

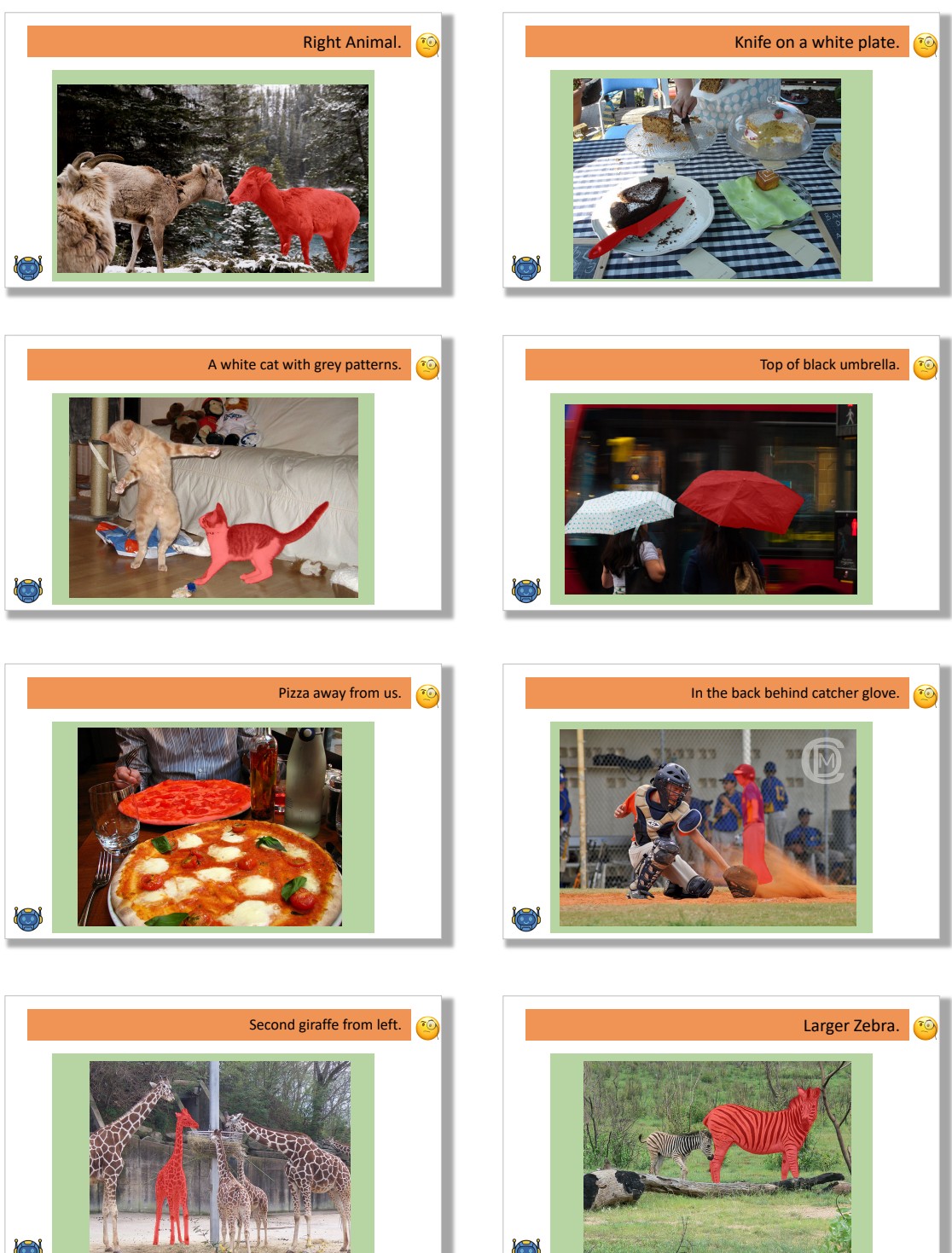

*Figure 8.* Visualization Results of Referring Segmentation. Visualized images are sampled from the RefCOCO Val set. FlowSeg with Boundary-Aware Refinement produces masks with cleaner boundaries.

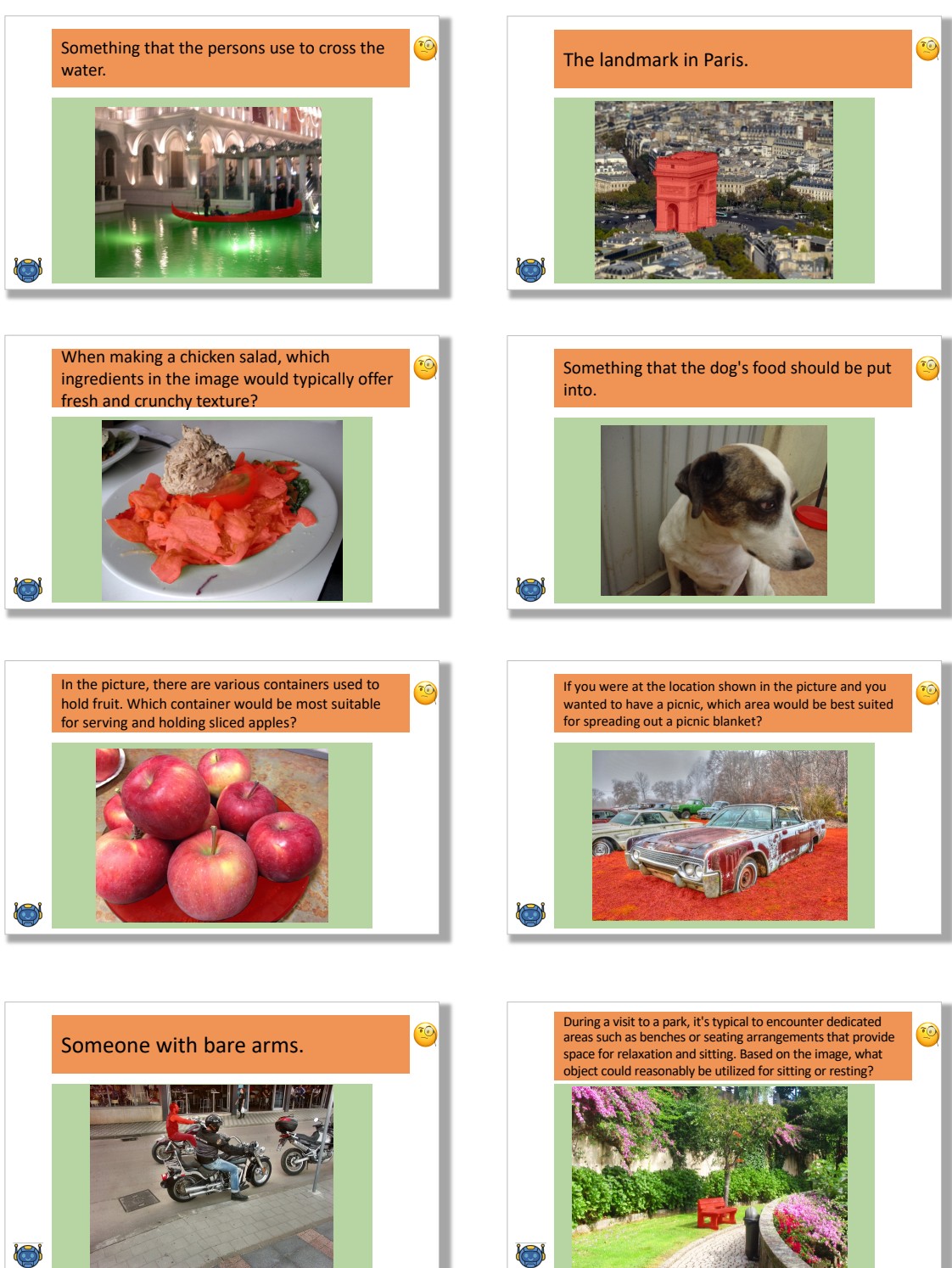

*Figure 9.* Visualization Results of Reasoning Segmentation. Visualized images are sampled from the reasoning segmentation Val set. FlowSeg with Boundary-Aware Refinement produces more accurate boundaries for complex reasoning tasks.

