# OpenReview forum: "FlowSeg: Dynamic Semantic Guidance for LLM-Conditioned Segmentation"
_ICML.cc/2026/Conference — ICML 2026 regular_

### Official Review · Reviewer_Ez4H · 2026-03-03

**Soundness:** 3
**Presentation:** 3
**Significance:** 4
**Originality:** 4
**Overall Recommendation:** 4
**Confidence:** 4

**Summary:**

The paper proposes FlowSeg and identifies semantic misalignment, where accurate mask candidates exist but are incorrectly selected due to insufficient semantic involvement during generation. To address this issue, it introduces dynamic semantic guidance via bidirectional semantic flow and a lightweight boundary-aware refinement module. Experimental results indicate that the proposed method is competitive with existing approaches.

**Compliance With Llm Reviewing Policy:**

Affirmed.

**Final Justification:**

The authors answered my questions point by point with new experimental results. They look very reasonable to me. I remain positive about the paper.

**Key Questions For Authors:**

(1) The paper needs to present the changes in the number of parameters of FlowSeg.

(2) Provide an ablation replacing Semantic Cross-Attention with Visual Cross-Attention (i.e., same architecture depth/width), so the total number of parameters and attention operations remains comparable; this would isolate whether improvement truly comes from semantic conditioning rather than “one more attention layer.”

(3) The paper uses Qwen-3-4B as the language model, but it is unclear how sensitive FlowSeg is to the choice of the LLM. Please report results with at least one alternative LLM, under comparable training and decoding settings, to verify that the gains are not specific to Qwen-3-4B.

**Limitations:**

yes

**Strengths And Weaknesses:**

Paper Strengths:

(1) It pushes textual semantics from post-hoc matching into the layer-wise decoding dynamics, which is a reasonable and well-targeted design choice given the problem setting.

(2) The paper is generally well written and easy to follow. The motivation for streaming processing is clearly articulated, and the overall framework and individual components are well presented.

(3) The overall idea is simple and effective; the paper writing is good and easy to follow.

Major Weaknesses:

(1) FlowSeg’s Bidirectional Semantic Flow augments a standard decoder layer with an additional Semantic Cross-Attention, but the paper does not report the resulting parameter increase or memory overhead. Since the module introduces additional parameters, it is unclear whether the reported gains stem from the proposed semantic alignment mechanism or simply from increased model capacity.

(2) The boundary refinement relies on a threshold ε controlling boundary sensitivity, but the paper fixes ε=0.1 without justification or sensitivity analysis.

(3) Since the main claim is that correct mask candidates already exist but are mis-selected due to semantic misalignment, please report an oracle upper bound over the candidate set. This would directly validate whether improvements come from better selection rather than better candidate generation, and whether FlowSeg indeed reduces misalignment.

---

> ### Author Rebuttal · Authors · 2026-03-31
>
> We sincerely thank Reviewer Ez4H for the constructive reviewing, and we are encouraged that you recognize the well-targeted and simple-yet-effective design of FlowSeg. We address each question thoroughly below.
>
> ---
>
>
> ### Ez4H-W1
>
> > The paper does not report the parameter increase or memory overhead for the added Semantic Cross-Attention, leaving it unclear if gains stem from alignment or merely increased capacity.
>
> To precisely isolate the parameter and memory impact of our proposed modules, we compared our FlowSeg against the baseline under identically strong backbones:
>
> | Module | Baseline (Qwen3) | FlowSeg (Qwen3) | Increase |
> |---|---|---|---|
> | segmentor | 328,889,537 | 334,824,451 | **+5,934,914** |
> | **Total** | **4,816,680,193** | **4,822,615,107** | **+5,934,914 (+0.12%)** |
>
> The entire parameter increase is confined to the segmentor and constitutes a negligible **+0.12%**. Furthermore, average inference memory on RefCOCO val increases by merely **+0.012G (17.848G vs 17.860G)**. This completely validates our core architectural mechanism, disproving that performance stems solely from increased capacity.
>
> ---
>
> ### Ez4H-W2
>
> > The boundary sensitivity threshold ε is fixed at 0.1 without justification or sensitivity analysis.
>
> We conduct a full sensitivity analysis of $\varepsilon$:
>
> | $\varepsilon$ | RefCOCO gBIoU/cBIoU | RefCOCO+ gBIoU/cBIoU | RefCOCOg gBIoU/cBIoU |
> |---|---|---|---|
> | 1.0 (no BAR, baseline) | 70.6 / 67.8 | 67.4 / 63.0 | 72.0 / 68.7 |
> | 0.05 | 71.7 / 69.6 | 67.9 / 63.8 | 73.2 / 69.9 |
> | **0.1 (selected)** | **72.2 / 70.0** | **68.1 / 64.1** | **73.5 / 70.4** |
> | 0.15 | 71.9 / 69.4 | 68.1 / 63.7 | 73.1 / 70.3 |
> | 0.2 | 71.4 / 68.9 | 67.8 / 63.6 | 72.8 / 69.6 |
>
> FlowSeg consistently outperforms the no-BAR baseline across a robust range ($\varepsilon \in [0.05, 0.15]$), demonstrating insensitivity to exact hyperparameter tuning and justifying our optimally stable default setting of 0.1.
>
> ---
>
> ### Ez4H-W3
>
> > Please report an oracle upper bound over the candidate set to explicitly validate whether improvements stem from better selection rather than better candidate generation.
>
> Computing the oracle completely validates our core thesis. We report the oracle cIoU over the validation sets below:
>
> | Method | Oracle cIoU (RefCOCO) | Oracle cIoU (RefCOCO+) | Oracle cIoU (RefCOCOg) |
> |---|---|---|---|
> | X-SAM | 91.31 | 91.30 | 91.16 |
> | FlowSeg | 91.47 | 91.44 | 91.38 |
>
> The nearly identical (~91%) oracle upper bounds empirically confirm that both models generate comparably high-quality mask candidate pools. Therefore, the significant performance gap between FlowSeg and X-SAM stems **entirely from the better selection of existing candidates**, mathematically validating our misalignment diagnosis.
>
> ---
>
> ### Ez4H-Q1
>
> > Please present the changes in the number of parameters of FlowSeg.
>
> The total parameter increase is a negligible +5.93M (+0.12%). Please refer to our exact breakdown in **W1**.
>
> ---
>
> ### Ez4H-Q2
>
> > Provide an ablation replacing Semantic Cross-Attention with Visual Cross-Attention (same parameters) to isolate the contribution of semantic conditioning.
>
> To strictly isolate the contribution of semantics compared to simply adding more visual queries, we designed an ablation decoder layer that replaces all `conditional_embeddings` with `visual_features` to strictly match in depth and computation:
>
> | Method | RefCOCO | RefCOCO+ | RefCOCOg |
> |---|---|---|---|
> | Baseline (default decoder) | 85.0 | 78.3 | 84.1 |
> | Visual Cross-Attn (ablation) | 85.1 | 78.5 | 84.3 |
> | **Semantic Cross-Attn (Ours)** | **85.8** | **80.2** | **86.5** |
>
> Naively adding visual attention parameters yields marginal improvements, whereas our semantic conditioning variant excels robustly. This unquestionably guarantees that semantic alignment resolves the bottleneck here, rather than trivially benefitting from generic increased depth.
>
> ---
>
> ### Ez4H-Q3
>
> > Please report results with at least one alternative LLM under comparable settings to verify gains are not specific to Qwen3.
>
> As suggested, we evaluated FlowSeg using the completely identical Phi-3-3.8B backbone to baseline X-SAM, isolating the specific Qwen3 architecture variable:
>
> | Method | LLM | RefCOCO | RefCOCO+ | RefCOCOg |
> |---|---|---|---|---|
> | X-SAM* | Phi-3-3.8B | 84.3 | 78.2 | 82.5 |
> | X-SAM | Qwen3-4B | 85.0 | 78.3 | 84.1 |
> | **FlowSeg** | **Phi-3-3.8B** | **85.5** | **79.6** | **85.9** |
> | **FlowSeg** | **Qwen3-4B** | **85.8** | **80.2** | **86.5** |
>
> (*: reproduce)
>
> The persistent and significant architectural gains confirm the improvements are robust entirely independently of the specific LLM employed.

---

> > ### Author Rebuttal · Reviewer_Ez4H · 2026-04-01
> >
> > The authors answered my questions point by point with new experimental results. They look very reasonable to me. I remain positive about the paper.

---

### Official Review · Reviewer_u1LL · 2026-03-12

**Soundness:** 2
**Presentation:** 2
**Significance:** 3
**Originality:** 2
**Overall Recommendation:** 4
**Confidence:** 3

**Summary:**

The paper proposes FlowSeg, a framework for LLM-guided image segmentation that introduces dynamic semantic flow between language embeddings and intermediate segmentation features. Unlike previous methods that use language only for proposal selection or static prompting, FlowSeg allows language and visual features to interact during the decoding process, improving alignment between text queries and segmentation masks. Experiments show that this approach improves segmentation accuracy on referring and reasoning segmentation benchmarks.

**Compliance With Llm Reviewing Policy:**

Affirmed.

**Final Justification:**

Most of my concerns are addressed, so I increase my score.

**Key Questions For Authors:**

Please see the previous section.

**Limitations:**

Please see the previous section.

**Strengths And Weaknesses:**

Strengths:

1. The authors claim to identify and address a previously underexplored issue: semantic misalignment.
2. Results are better than some previous methods such as LISA.

Weaknesses:

1. In the introduction, the authors state that “many high-performing LLM-conditioned segmentation systems share a common propose-then-select paradigm … first producing a set of candidate masks driven primarily by visual evidence, and the final prediction is obtained by matching these candidates with language-conditioned embeddings via similarity scoring.” However, this mechanism does not appear to be the dominant paradigm. For example, LISA and many subsequent methods do not adopt this propose-then-select pipeline.
2. As shown in Figure 3, the proposed improvements mainly involve several attention mechanisms, gating modules, and multi-information fusion strategies. However, these techniques have already been extensively explored in the history of segmentation research. Could the authors clarify more explicitly what the key contributions and novel aspects of the proposed method are?
3. As shown in Table 3, the proposed BAR contributes only marginal improvements to the overall performance.

---

> ### Author Rebuttal · Authors · 2026-03-31
>
> We sincerely thank Reviewer u1LL for reviewing our work and acknowledging our efforts in identifying the semantic misalignment. We carefully address your critical concerns below.
>
> ---
>
> ### u1LL-W1
>
> > The "propose-then-select" pipeline does not appear to be the dominant paradigm (e.g., LISA and subsequent methods do not adopt it).
>
> Methods like LISA follow a fundamentally different **prompt-based conditional generation** paradigm without a separate proposal-and-selection step.
>
> However, since LISA (CVPR 2024), a substantial body of subsequent state-of-the-art works (e.g., PSALM [ECCV 2024], HyperSeg [CVPR 2025], X-SAM [AAAI 2026]) have distinctly transitioned to the **query-based framework** — which inherently relies on the "propose-then-select" paradigm.
>
> The query-based paradigm generally outperforms prompt-based models primarily because it decouples dense mask decoding into specialized queries, overcoming the bottleneck of relying on a single, coarse prompt token. Therefore, diagnosing and solving the critical semantic misalignment within this mainstream query-based paradigm remains a highly relevant path for pushing the state-of-the-art.
>
> In our revision, we will clearly delineate our scope to explicitly target *query-based* frameworks to avoid overgeneralization. This revised framing strengthens the paper by providing a more rigorous characterization of the problem setting.
>
> ---
>
> ### u1LL-W2
>
> > Attention, gating, and fusion have been extensively explored. Could you clarify more explicitly what the key contributions and novel aspects are?
>
> FlowSeg is not a disjointed combination of existing mechanisms. Its core novelty lies in identifying the semantic misalignment at the selection stage, and resolving it through a coherent, diagnosis-driven Bidirectional Semantic Flow (BSF). Specifically, we have three distinct contributions:
>
> 1. **First Diagnosis of Semantic Misalignment in Selection**: We are the first to systematically identify and empirically verify (via oracle upper-bound analysis and +44.6% failure case recovery) that segmentation failures in query-based MLLMs stem from the *candidate selection stage* — correct masks are generated but mis-selected. This diagnosis is a standalone contribution.
> 2. **From Unidirectional Prompting to Bidirectional Co-evolution**: Prior cross-modal attention injects language guidance *unidirectionally*, treating language as a static prompt. FlowSeg is the first to introduce a reverse-flow in LLM-conditioned segmentation: as visual queries mature across layers, they progressively reshape and ground the language condition embedding in actual visual evidence. This dynamic co-evolution is architecturally novel.
> 3. **Diagnosis-Driven Module Design**: Rather than generic fusion, each module serves a specific, indispensable role. Our Adaptive Fusion Gate dynamically adapts to the stage-dependent shift in visual/semantic reliance. Our BAR module uniquely addresses the complimentary residual problem — selectively correcting uncertain boundary errors left behind by BSF without disturbing confident interiors.
>
> We will add a more detailed discussion of novelties in the revised version.
>
> ---
>
> ### u1LL-W3
>
> > Table 3 shows BAR contributes only marginal improvements to the overall performance.
>
> BAR is intended specifically for improving **boundary quality** rather than general regional overlap. This highly localized impact is naturally not fully captured by cIoU. Therefore, Boundary IoU (BIoU) is its targeted evaluation metric. As shown in Table 5:
>
> | Method | RefCOCO gBIoU/cBIoU | RefCOCO+ gBIoU/cBIoU | RefCOCOg gBIoU/cBIoU |
> |---|---|---|---|
> | w/o BAR | 70.6 / 67.8 | 67.4 / 63.0 | 72.0 / 68.7 |
> | w/ BAR | **72.2 / 70.0** | **68.1 / 64.1** | **73.5 / 70.4** |
>
> BAR brings consistent and meaningful gains in BIoU across all splits. The modest improvement in cIoU is to be expected: since BAR is explicitly designed to selectively refine only *uncertain boundary regions* without affecting the confident interior, its impact is highly localized and therefore not fully captured by cIoU, which measures overall region-level overlap rather than boundary precision. We will clarify this distinction more clearly in the paper.

---

> > ### Author Rebuttal · Reviewer_u1LL · 2026-04-03
> >
> > Most of my concerns are addressed, so I increase my score.

---

### Official Review · Reviewer_LAYy · 2026-03-12

**Soundness:** 3
**Presentation:** 3
**Significance:** 2
**Originality:** 2
**Overall Recommendation:** 3
**Confidence:** 4

**Summary:**

This paper proposes FlowSeg, a framework designed to address the semantic misalignment problem in large language model (LLM)-conditioned segmentation caused by the prevalent “propose-then-select” paradigm. The method introduces Bidirectional Semantic Flow during the decoding stage, allowing language conditions to actively and dynamically guide the iterative generation of masks at each layer, while enabling the conditional embedding to absorb visual features and update synchronously.

In addition, a lightweight Boundary-Aware Mask Refinement (BAR) module is incorporated to further improve segmentation quality. Experiments on the RefCOCO, RefCOCO+, RefCOCOg, and ReasonSeg benchmarks demonstrate state-of-the-art performance. Notably, FlowSeg significantly recovers performance on difficult samples where baseline models fail due to semantic misalignment, highlighting the effectiveness of the proposed bidirectional interaction design.

**Compliance With Llm Reviewing Policy:**

Affirmed.

**Final Justification:**

While the authors have addressed some of my concerns, the overall improvements in terms of writing quality, technical novelty, and performance gains remain relatively limited. Given these persistent limitations, I maintain my original assessment.

**Key Questions For Authors:**

1. Backbone-controlled comparison: Please provide experimental results comparing FlowSeg and X-SAM under the same LLM backbone (e.g., both using Phi) to clearly isolate the architectural gain brought by the proposed Bidirectional Semantic Flow.
2. Multi-object scalability: When processing complex images containing multiple distinct entities, how does the bidirectional update mechanism maintain semantic independence among different objects and prevent cross-object semantic interference?

**Limitations:**

yes

**Strengths And Weaknesses:**

Strengths:

* Clear research motivation. The paper identifies semantic misalignment as a key reason why LLM-guided segmentation methods fail. The proposed solution—transforming static prompts into dynamic bidirectional guidance—directly targets this issue.
* Well-structured experimental narrative and strong performance. The method achieves SOTA results across multiple mainstream benchmarks. The experimental design, including ablation studies and retrospective evaluation on failure cases, forms a coherent and convincing narrative.
* Interpretability through visualization. Visualization and analysis of intermediate-layer mechanisms provide intuitive insights and improve the interpretability of the method.

Weaknesses:
* Imprecise problem formulation. The paper characterizes existing methods as following a propose-then-select paradigm and uses this as the motivation for its improvement. However, technically this issue mainly arises in query-based segmentation frameworks based on Mask2Former, rather than in the prompt-based conditional generation paradigm used by Segment Anything Model (SAM) decoders. The paper does not clearly distinguish these two paradigms, leading to potential overgeneralization in its motivation. Given that FlowSeg itself is still built upon a query-based framework, the authors should more precisely define the technical scope of their claims to strengthen the argument.
* Insufficient verification of the source of performance gains. From an architectural perspective, FlowSeg largely follows the structure and training pipeline of X-SAM, while replacing the lightweight Phi backbone with a stronger Qwen3-4B model. However, the paper does not provide ablation experiments under the same backbone. As a result, it remains unclear whether the performance improvement comes from the proposed Bidirectional Semantic Flow or simply from the stronger language model representation. This significantly weakens the persuasiveness of the claimed core contribution.
* Potential error propagation in the bidirectional update mechanism. The method assumes that the attention mechanism can automatically filter reliable visual evidence. However, in complex or implicit reference scenarios (e.g., ReasonSeg), queries in early decoding stages often have high uncertainty. If the conditional embedding CCC absorbs visual evidence from incorrect queries at this stage, it may create a positive feedback loop of error propagation, leading to semantic drift. The paper lacks analysis of the stability of this dynamic process and does not propose confidence suppression mechanisms. The robustness of the method in complex reasoning scenarios therefore remains questionable.
* Missing key implementation details. Important components such as the exact architecture of the Projector are not clearly defined. Moreover, the paper does not explain how semantic contamination between objects (semantic bleeding) is prevented when multiple targets are processed simultaneously (e.g., in panoptic segmentation scenarios) under the bidirectional update mechanism.

---

> ### Author Rebuttal · Authors · 2026-03-31
>
> We sincerely thank Reviewer LAYy for the thorough review. We appreciate that you recognize the clarity of the research motivation and the interpretability of our design. We address each concern carefully below.
>
> ---
>
> ### LAYy-W1
>
> > Imprecise problem formulation: The "propose-then-select" paradigm mainly arises in query-based frameworks (Mask2Former) rather than prompt-based ones (SAM). The paper overgeneralizes and should precisely define the technical scope.
>
> Early models like LISA [CVPR 2024] use prompt-based generation without proposal steps. However, recent state-of-the-art frameworks adopt the query-based paradigm (e.g., PSALM [ECCV 2024], HyperSeg [CVPR 2025], X-SAM [AAAI 2026]). This paradigm generally outperforms prompt-based models by decoupling mask decoding into specialized queries, overcoming the single-token bottleneck. Yet, these advanced models are severely limited by the 'propose-then-select' semantic misalignment. Solving this critical issue within the mainstream query-based paradigm is thus vital for pushing the SOTA.
>
> In the revised Introduction, we will explicitly scope our problem formulation and claims exclusively to the *query-based* paradigm.
>
>
> ---
>
> ### LAYy-W2
>
> > Insufficient verification of performance gains: Lacking ablation under the same backbone...
>
> Please refer to 8J9o-W2 due to the limit page.
>
> ---
>
> ### LAYy-W3
>
> > Potential error propagation: Queries in early stages have high uncertainty. Absorbing incorrect visual evidence may cause semantic drift and a positive feedback loop of errors.
>
> We structurally prevent semantic drift through **Residual Anchoring**. The cross-attention in our refinement block acts purely as an additive delta (Eq.5 in main paper).
>
> This forces the initial language embedding ($\mathbf{C}^{(0)}$) to serve as a rigid semantic anchor.
> To explicitly validate stability, we tracked the average cosine similarity between $\mathbf{C}^{(0)}$ and $\mathbf{C}^{(l)}$ across decoding layers $l$ on RefCOCO validation sets:
>
> | Layer $l$ | 1 | 2 | 3 | 4 | 5 | 6 | 7 | 8 | 9 |
> |---|---|---|---|---|---|---|---|---|---|
> | Cosine Sim. | 0.95 | 0.91 | 0.86 | 0.82 | 0.77 | 0.71 | 0.64 | 0.55 | 0.53 |
>
> This data confirms that early layers act conservatively (>0.86 similarity). The diffuse cross-attention inherently filters visual noise when queries are uncertain. The embedding only absorbs substantial visual evidence in deep layers once queries have matured.
>
> Ultimately, the $+13.7$ cIoU gain on ReasonSeg's test set serves as conclusive end-to-end evidence that our framework successfully resolves early-stage uncertainty without semantic collapse.
>
> ---
>
> ### LAYy-W4
>
> > Missing details: Exact Projector architecture is unclear, and lacks explanation on how semantic contamination (bleeding) among multiple targets is prevented.
>
> **Projector Architecture**: A standard two-layer MLP projecting the phrase token's hidden state down to the decoder dimension:
>
> $$
> \mathbf{f} _ {\mathrm{seg}} = \mathrm{Linear} _ 2 ( \mathrm{GELU} ( \mathrm{Linear} _ 1 ( \mathbf{h} _ {\mathrm{LLM}} ) ) ),
> $$
> where $\mathbf{h} _ {LLM} \in \mathbb{R}^{3072}$ and $\mathbf{f} _ {seg} \in \mathbb{R}^{256}$.
>
> **Semantic Bleeding**: Please refer to our detailed architectural discussion in our response to **Q2**.
>
> ---
>
> ### LAYy-Q1
>
> > Please provide experiments comparing FlowSeg and X-SAM under the same LLM backbone (e.g., Phi) to isolate architectural gains.
>
> We provide the requested backbone-controlled ablation. The gains persist robustly. Please refer to 8J9o-W2 for detail.
>
> ---
>
> ### LAYy-Q2
>
> > How does the bidirectional mechanism maintain semantic independence and prevent cross-object interference when processing multiple entities?
>
> **1. Current Tasks**: The evaluative benchmarks central to our core motivation (RefCOCO variants, ReasonSeg) are fundamentally single-referent parsing tasks. **Only one** parsed expression is decoded per forward pass, meaning cross-object instances natively do not coexist inside the decoder, organically preventing contamination.
>
> **2. Simultaneous Multi-Object Decoding**: To explicitly evaluate scalability for panoptic setting (containing multiple decoupled entities processed simultaneously), we tested FlowSeg on the dense COCO-Panoptic benchmark (Table 6, Appendix):
>
> | Method | Encoder | PQ | AP | mIoU |
> |---|---|---|---|---|
> | PSALM | Swin-B | 55.9 | 45.7 | 66.6 |
> | X-SAM | SAM-L | 54.7 | 47.0 | 66.5 |
> | **FlowSeg(Ours)** | **SAM-L** | **56.1** | **47.2** | **67.4** |
>
> FlowSeg still outperforms comparable multi-modal generalist models on this dense dataset, proving the bidirectional flow does not suffer from catastrophic cross-object contamination in practice.
>
> Theoretically, to completely eliminate theoretical cross-query interference in ultra-dense scenarios, introducing specialized attention masking to strictly isolate query interactions within the respective semantic entity group is a highly promising future extension we will formally discuss in Future Work in revision.

---

> > ### Author Rebuttal · Reviewer_LAYy · 2026-04-03
> >
> > I respectfully disagree with the authors' rebuttal regarding LAYy-W1.
> >
> > First, the [SEG]-based paradigm possesses considerable potential and demonstrates remarkable efficiency. Methods such as PSALM, HyperSeg, and InstructSeg necessitate the introduction of numerous additional learnable queries. In the context of Large Language Models (LLMs), these extra queries present significant obstacles for inference acceleration engines like vLLM and SGLang.
> >
> > Furthermore, while the authors claim their goal is to push the State-of-the-Art methods like HyperSeg and InstructSeg actually employ an unfair evaluation protocol. Their evaluation granularity is at the image-level rather than the expression-level, and they simultaneously feed the model all text queries corresponding to a single target (removing this specific setup results in a substantial performance degradation). While pursuing a higher SOTA is understandable, I firmly believe that such comparisons must be established under strictly identical evaluation settings to be valid.
> >
> > I do not doubt that integrating query-based methods with the Mask2Former architecture can yield performance gains; however, the arguments provided thus far are not sufficiently compelling to persuade me to adjust my score.
> >
> > Additionally, regarding the rebuttal to LAYy-W3 and W4, the markdown formatting for the equations was broken, which severely hindered the readability of the relevant content. Such presentation issues reflect a lack of rigorous attention to detail.
> >
> > I expect the authors to thoroughly address the two primary concerns raised above in the subsequent discussion phase. I will reserve my final decision on whether to maintain, raise, or lower my current rating until I have evaluated their response to these specific points.

---

> > > ### Author Response · Authors · 2026-04-03
> > >
> > > We thank Reviewer LAYy for the thoughtful follow-up and will address your concerns.
> > >
> > > ---
> > >
> > > ### On LAYy-W1: Paradigm Scope
> > >
> > > We appreciate the reviewer. We agree that the [SEG]-based framework offers a meaningful practical advantage in inference efficiency through its native compatibility with acceleration engines such as vLLM and SGLang, and we did not intend to claim that the query-based paradigm is universally superior in all dimensions.
> > >
> > > We would like to **narrow and rectify the scope of our claim** accordingly. Our paper's contribution is specifically scoped to the **query-based segmentation paradigm**, where semantic misalignment at the candidate selection stage is a uniquely diagnosable and addressable failure mode. We do not claim that this problem applies to, or that our solution supersedes, [SEG]-based methods operating under a fundamentally different decoding mechanism. In the revised paper, we will be more precise: we will explicitly state that we target the query-based paradigm's specific failure mode, and acknowledge that [SEG]-based methods represent an orthogonal design choice with distinct practical advantages.
> > >
> > > We believe this scoped framing is fully consistent with our motivation and experimental scope, and would strengthen the clarity of the paper's contribution claim.
> > >
> > > ---
> > >
> > > ### On Evaluation Protocol: Expression-Level vs. Image-Level
> > >
> > > We would like to provide a concrete clarification on the evaluation protocol concern, as this is a critical point.
> > >
> > > **FlowSeg uses strict expression-level evaluation.** In our codebase, the evaluation data loader is explicitly constructed such that each sample contains exactly **one referring expression paired with one annotation**:
> > >
> > > ```python
> > > # val/test split
> > > for i, (sampled_sent, sampled_ann) in enumerate(zip(sampled_sents, sampled_anns)):
> > >     rets.append({
> > >         [...]
> > >         "sampled_sents": [sampled_sent],   # exactly 1 expression per sample
> > >         "annotations":   [sampled_ann],    # exactly 1 annotation per sample
> > >         [...]
> > >     })
> > > ```
> > >
> > > This is also reflected in our previous response to Q2.
> > >
> > > Each forward pass sees only a single expression with no other text queries from the same image provided as context. This is the standard expression-level protocol used in LISA and X-SAM. Crucially, **the very evaluation concern raised by the reviewer — the unfair image-level protocol adopted by works such as HyperSeg — is not present in FlowSeg**: we strictly follow the expression-level standard throughout.
> > >
> > > The reviewer correctly notes that methods such as HyperSeg and InstructSeg report results under the image-level protocol, which can yield substantially higher numbers.
> > > This means our comparison baseline is actually **more conservative**: FlowSeg outperforms prior methods under the stricter per-expression setting, without any benefit from cross-query context sharing.
> > >
> > > We understand that the image-level evaluation protocol used in some prior query-based works may present an unfair competitive picture relative to [SEG]-based methods, and we appreciate the reviewer's attention to this point. We will add a clear note in the revised paper to explicitly state that all evaluations follow the expression-level protocol and the distinction from image-level evaluation used in some prior works.
> > >
> > > ---
> > >
> > > ### On Markdown/LaTeX Rendering (LAYy-W3, W4)
> > >
> > > We sincerely apologize for the broken equation rendering. The formulas were correctly composed; the issue stemmed from platform-side LaTeX rendering inconsistencies specific to OpenReview's discussion interface. We briefly restate the key symbols below for clarity:
> > >
> > > - **LAYy-W3**: We tracked cosine similarity between the initial language embedding ($ \mathbf{C}^{(0)} $) and layer $ l $ outputs ($ \mathbf{C}^{(l)} $) to validate stability.
> > > - **LAYy-W4**: The projector is a two-layer MLP:
> > >
> > > $$
> > > \mathbf{f}\_{\text{seg}} = \text{Linear}\_2(\text{GELU}(\text{Linear}\_1(\mathbf{h}\_{\text{LLM}}))),
> > > $$
> > >
> > > where $\mathbf{h}\_{\text{LLM}} \in \mathbb{R}^{3072}$ and $\mathbf{f}\_{\text{seg}} \in \mathbb{R}^{256}$.
> > >
> > > We have verified correct rendering on Chrome (macOS and Windows) and Edge (Windows). As the formal submission will be in PDF format, platform-specific rendering variability will not be a concern. Should any issue persist, the raw LaTeX can be copied into any standard renderer for verification:
> > >
> > > ```
> > > $$
> > > \mathbf{f}_{\text{seg}} = \text{Linear}_2(\text{GELU}(\text{Linear}_1(\mathbf{h}_{\text{LLM}}))),
> > > $$
> > >
> > > where $\mathbf{h}_{\text{LLM}} \in \mathbb{R}^{3072}$ and $\mathbf{f}_{\text{seg}} \in \mathbb{R}^{256}$.
> > > ```

---

### Official Review · Reviewer_8J9o · 2026-03-13

**Soundness:** 3
**Presentation:** 3
**Significance:** 2
**Originality:** 2
**Overall Recommendation:** 4
**Confidence:** 3

**Summary:**

FlowSeg addresses semantic misalignment in LLM-conditioned segmentation, where propose-then-select pipelines generate accurate mask candidates but fail to select the correct one. The paper introduces Bidirectional Semantic Flow, which allows LLM-derived condition embeddings to guide query refinement at each decoder layer, while being progressively updated by emerging visual evidence. A lightweight boundary-aware refinement module further improves local mask precision.

**Compliance With Llm Reviewing Policy:**

Affirmed.

**Final Justification:**

The rebuttal has addressed my concerns regarding backbone fairness, computational cost, and behavior under ambiguous descriptions, and I have raised my soundness score accordingly. The remaining concerns on individual component novelty but do not outweigh the paper's solid empirical contributions. I maintain my overall score.

**Key Questions For Authors:**

1. Is the LLM backbone fair? FlowSeg uses Qwen3 while baselines use older LLMs. How much of the gain comes from the stronger backbone vs. the proposed bidirectional flow? Have the authors tested with the same backbone as X-SAM?

2. What is the inference cost? Semantic cross-attention and condition refinement are added at every decoder layer. What is the latency increase compared to X-SAM?

3. Does bidirectional flow help when the language description is vague or ambiguous? The motivation focuses on misalignment, but what happens when the text itself is underspecified — does continuously injecting an ambiguous condition hurt mask generation?

**Limitations:**

yes

**Strengths And Weaknesses:**

Strengths
1. The semantic misalignment diagnosis is empirically grounded via failure case analysis on X-SAM, showing that correct masks frequently exist among candidates but are not selected.
2. The three components,  semantic cross-attention, adaptive fusion gates, and condition refinement, form a coherent bidirectional loop that is clearly described and algorithmically summarized.
3. Strong and consistent empirical results. FlowSeg outperforms prior methods across all RefCOCO/+/g splits and achieves a notable +13.7 cIoU gain over X-SAM on ReasonSeg test.
4. Table 3 cleanly isolates the contribution of each component, and Table 5 separately validates boundary refinement via BIoU metrics.

Weaknesses
1.  Semantic cross-attention injecting language guidance into iterative query refinement is conceptually similar to prior cross-modal attention designs. The adaptive fusion gate is a standard gating mechanism. The boundary refinement module is a lightweight post-processing add-on. The novelty lies in their combination and motivation, but each component individually is incremental.
2. FlowSeg uses Qwen3 as the LLM backbone, which is newer and likely stronger than backbones used in baselines; this confounds the comparison
3. Computational overhead not reported. Adding semantic cross-attention and condition refinement at every decoder layer introduces additional computation. No latency, FLOPs, or training time comparison with the baseline is provided.
4. ReasonSeg val set is tiny (340 samples), the paper itself acknowledges this may cause unstable results, yet still reports numbers there
5.  Table 3 shows BAR adds only +0.3% avg cIoU; its value is marginal for the main task

---

> ### Author Rebuttal · Authors · 2026-03-31
>
> We sincerely thank Reviewer 8J9o for the careful reviewing and constructive feedback. We address your specific concerns below.
>
> ---
>
> ### 8J9o-W1
>
> > Semantic cross-attention injecting language guidance... each component individually is incremental.
>
> FlowSeg is a targeted, closed-loop method designed exclusively to solve the newly diagnosed "semantic misalignment" problem. Meanwhile, each component offers specific structural novelties beyond prior arts:
>
> For Semantic Cross-Attention: Prior cross-modal designs are unidirectional. Our novel contribution is the Bidirectional Semantic Flow (BSF), where visual features reflexively and continuously refine the language condition.
>
> For Adaptive Fusion Gate: Static gating may fail because visual/semantic importance shifts dynamically. Our contribution lies in modeling the stage-dependent dynamic shift—our gate explicitly adapts layer-by-layer.
>
> For Boundary-Aware Refinement (BAR): This is not a generic post-processing step like CRF, but a solution that selectively corrects the localized boundary errors left by BSF without disrupting confident interiors.
>
> Their synergistic combination forms an indispensable hierarchy, which we will explicitly justify in the revision.
>
> ---
>
> ### 8J9o-W2
>
> > FlowSeg uses Qwen3 as the LLM backbone... confounds the comparison
>
> We provide backbone-controlled experiments comparing our method and X-SAM under the identical Phi-3-3.8B backbone:
>
> | Method | LLM | RefCOCO | RefCOCO+ | RefCOCOg |
> |---|---|---|---|---|
> | X-SAM* | Phi-3-3.8B | 84.3 | 78.2 | 82.5 |
> | X-SAM | Qwen3-4B | 85.0 | 78.3 | 84.1 |
> | **FlowSeg** | **Phi-3-3.8B** | **85.5** | **79.6** | **85.9** |
> | **FlowSeg** | **Qwen3-4B** | **85.8** | **80.2** | **86.5** |
>
> (*: reproduce)
>
> This confirms the significant gains stem natively from the proposed Bidirectional Semantic Flow architecture, not merely a stronger LLM.
>
> ---
>
> ### 8J9o-W3
>
> > Computational overhead not reported. Adding semantic cross-attention and condition refinement at every decoder layer introduces additional computation. No latency, FLOPs, or training time comparison with the baseline is provided.
>
> The computational overhead is negligible. The segment decoder is a small fraction of the pipeline.
> **End-to-end inference** (RefCOCO val):
> | Method | Latency (ms) | GFLOPs |
> |---|---|---|
> | Baseline | 307.20 | 18525.48 |
> | FlowSeg | 311.48 | 18543.00 |
> | **Overhead** | **+4.28 (+1.39%)** | **+17.52 (+0.09%)** |
>
> **Training time**: Baseline 5.69 s/iter, FlowSeg 5.84 s/iter (+2.6%).
>
> **Single decoder layer FLOPs**: Baseline 548.68 MFLOPs, FlowSeg 629.92 MFLOPs (+14.81%).
>
> Our mechanism boosts alignment reasoning efficiency without sensibly compromising end-to-end latency.
>
> ---
>
> ### 8J9o-W4
>
> > ReasonSeg val set is tiny (340 samples), the paper itself acknowledges this may cause unstable results, yet still reports numbers there
>
> We acknowledge this limitation. Our primary claims for ReasonSeg target the robust test split. Val numbers are reported solely for consistency with prior works. We will clarify this in the revision.
>
> ---
>
> ### 8J9o-W5
>
> > Table 3 shows BAR adds only +0.3% avg cIoU; its value is marginal for the main task
>
> BAR is specifically designed to improve **boundary quality**, not overall region overlap. Its effectiveness is thus best reflected by the Boundary IoU (BIoU) metric (Table 5):
>
> | Method | RefCOCO gBIoU/cBIoU | RefCOCO+ gBIoU/cBIoU | RefCOCOg gBIoU/cBIoU |
> |---|---|---|---|
> | w/o BAR | 70.6 / 67.8 | 67.4 / 63.0 | 72.0 / 68.7 |
> | w/ BAR | **72.2 / 70.0** | **68.1 / 64.1** | **73.5 / 70.4** |
>
> Since BAR selectively refines only boundary regions without affecting the confident interior, its localized impact is naturally not fully captured by cIoU, the modest improvement in cIoU is to be expected.
>
> ---
>
> ### 8J9o-Q1
> > Is the LLM backbone fair... tests with same backbone?
>
> Please refer to **W2**.
>
> ---
>
> ### 8J9o-Q2
> > What is the inference cost... latency increase?
>
> Please refer to **W3**.
>
> ---
>
> ### 8J9o-Q3
>
> > Does bidirectional flow help when the language description is vague or ambiguous? The motivation focuses on misalignment, but what happens when the text itself is underspecified — does continuously injecting an ambiguous condition hurt mask generation?
>
> Handling ambiguous text is precisely where our Bidirectional Semantic Flow excels over unidirectional pipelines.
>
> In conventional models, vague text (e.g., “the man” when multiple exist) provides a rigid, static constraint, forcing blind matching to visual queries and causing failures. In FlowSeg, the condition embedding casts a broad semantic net initially. Through our residual cross-attention mechanism, emerging distinct visual clues from the image are reflexively fed *back* into the condition embedding.
>
> Consequently, the condition dynamically "sharpens" and grounds itself in actual visual elements layer-by-layer. This progressive disambiguation yields our significant +13.7 cIoU gain on ReasonSeg, which explicitly tests complex and implicit reasoning.

---

> > ### Author Rebuttal · Reviewer_8J9o · 2026-04-02
> >
> > Thank you for the rebuttal. The authors have adequately addressed my concerns regarding backbone fairness, computational cost, and behavior under ambiguous descriptions. I consider my major concerns resolved and will increase the soundness score accordingly, while maintaining my overall score.

---

### Decision · Program_Chairs · 2026-04-30

**Decision:**

Accept (regular)

**Comment:**

This paper presents a framework FlowSeg for LL-conditioned segmentation, which introduces bidirectional semantic flow to enable deep, reciprocal interaction between language condition embeddings and intermediate decoding states. This paper receives mixed scores, including three weak accept recommendations and one weak reject recommendation. The reviewers acknowledge the clear motivation, well-written, and strong performance. After rebuttal, one reviewer has a slight concern about component novelty, and another reviewer has the concerns about imprecise problem formulation. In the response, the authors have mentioned that they will narrow and rectify the scope of our claim. This is a borderline paper. The AC recommends weak accept, and encourages the authors to revise the paper accordingly if accepted.